# Automated morphological phenotyping using learned shape descriptors and functional maps: A novel approach to geometric morphometrics

Oshane O. Thomas[1]*, Hongyu Shen[2], Ryan L. Raaum[3,4,5], William E. H. Harcourt-Smith[3,4,5,6], John D. Polk[1,7,8‡], Mark Hasegawa-Johnson[2‡]

1 Department of Anthropology, University of Illinois at Urbana-Champaign, Urbana, Illinois, United States of America, 2 Department of Electrical and Compute Engineering, University of Illinois at Urbana-Champaign, Urbana, Illinois, United States of America, 3 Department of Anthropology, Lehman College, Bronx, New York, United States of America, 4 Department of Anthropology, CUNY Graduate Center, New York, New York, United States of America, 5 New York Consortium in Evolutionary Primatology, New York, New York, United States of America, 6 Department of Vertebrate Paleontology, American Museum of Natural History, New York, New York, United States of America, 7 Department of Biomedical and Translational Sciences, University of Illinois at Urbana-Champaign, Urbana, Illinois, United States of America, 8 Department of Anthropology, University at Albany, Albany, New York, United States of America

‡ These authors are joint senior authors on this work.
* othomas2@illinois.edu

**Data Availability Statement:** Our code can now be found at https://github.com/oothomas/morphVQ,

## Abstract

The methods of geometric morphometrics are commonly used to quantify morphology in a broad range of biological sciences. The application of these methods to large datasets is constrained by manual landmark placement limiting the number of landmarks and introducing observer bias. To move the field forward, we need to automate morphological phenotyping in ways that capture comprehensive representations of morphological variation with minimal observer bias. Here, we present Morphological Variation Quantifier (morphVQ), a shape analysis pipeline for quantifying, analyzing, and exploring shape variation in the functional domain. morphVQ uses descriptor learning to estimate the functional correspondence between whole triangular meshes in lieu of landmark configurations. With functional maps between pairs of specimens in a dataset we can analyze and explore shape variation. morphVQ uses Consistent ZoomOut refinement to improve these functional maps and produce a new representation of shape variation, area-based and conformal (angular) latent shape space differences (LSSDs). We compare this new representation of shape variation to shape variables obtained via manual digitization and auto3DGM, an existing approach to automated morphological phenotyping. We find that LSSDs compare favorably to modern 3DGM and auto3DGM while being more computationally efficient. By characterizing whole surfaces, our method incorporates more morphological detail in shape analysis. We can classify known biological groupings, such as Genus affiliation with comparable accuracy. The shape spaces produced by our method are similar to those produced by modern 3DGM and to auto3DGM, and distinctiveness functions derived from LSSDs show us how shape variation differs between groups. morphVQ can capture shape in an automated fashion

and our data has been available at https://doi.org/10.5061/dryad.bvq83bkcr.

**Funding:** The study "Automated morphological phenotyping using learned shape descriptors and functional maps: A novel approach to geometric morphometrics" was done in part with support from the National Science Foundation (award number BCS-0962903) awarded to JDP. This funding supported the microCT scanning and digitization of the mouse humeri specimen used in the study. None of the authors received financial compensation or salary for this work.The funders had no role in study design, data collection and analysis, decision to publish, or preparation of the manuscript.

**Competing interests:** All authors declare that they have no conflicts of interest.

while avoiding the limitations of manually digitized landmarks, and thus represents a novel and computationally efficient addition to the geometric morphometrics toolkit.

## Author summary

The quantification of biological shape variation has relied on expert placement of relatively small subsets of landmarks and their analysis using tools of geometric morphometrics (GM). This paper introduces morphVQ, a novel, automated, learning-based approach to shape analysis that approximates the non-rigid correspondence between surface models of bone. With accurate functional correspondence between bones, we can characterize the shape variation within a dataset. Our results demonstrate that morphVQ performs similarly to manual digitization and to an existing automated phenotyping approach, auto3DGM. morphVQ has the advantages of greater computational efficiency and while capturing shape variation directly from surface model representations of bone. We can classify biological shapes to the Genus level with comparable accuracy to previous approaches, and we can demonstrate which aspects of bone shape differ most between groups. The ability to provide comparable accuracy in a Genus level classification with features extracted from morphVQ further guarantees the validity of this approach.

This is a *PLOS Computational Biology* Methods paper.

## 1 Introduction

Biologists studying bone surface' morphology often quantify shape using the landmarks and semilandmarks of Geometric Morphometrics (GM) [1–6]. Such quantification permits us to analyze the differences between bones with manually identified homologous or corresponding landmarks. With landmarks, we can study biological features' locations in geometric relation to each other, which provides opportunities to examine the patterning of complex morphological phenotypes [4, 7, 8]. Modern three-dimensional GM analysis pipelines begin with the manual collection of tens to hundreds of landmarks from digital representations of bone such as triangular meshes (polygon models). They end with a set of Procrustes-aligned coordinate shape variables that retain the geometric information contained within the data [5] (Fig 1). These shape variables, geometric measures of an object invariant to location, scale, and orientation, are the object of these analyses. They make it possible to ask research questions about the structuring of biological shape variation. Consequently, GM analyses of shape variation have increased in importance and remain indispensable in theoretical and mathematical biology [5, 8].

In implementing landmark-based GM, morphologists must make a host of decisions and compromises that limit the range of phenotypic variability one can capture [9]. Researchers must make informed sacrifices about which parts of morphology to study by choosing a fixed number of landmarks of various types to capture the geometric features we judge to be most important to our questions. Critically, we are required to decide what aspects of morphology must be measured and how to use landmarks to quantify that variation [9–11]. Landmarks

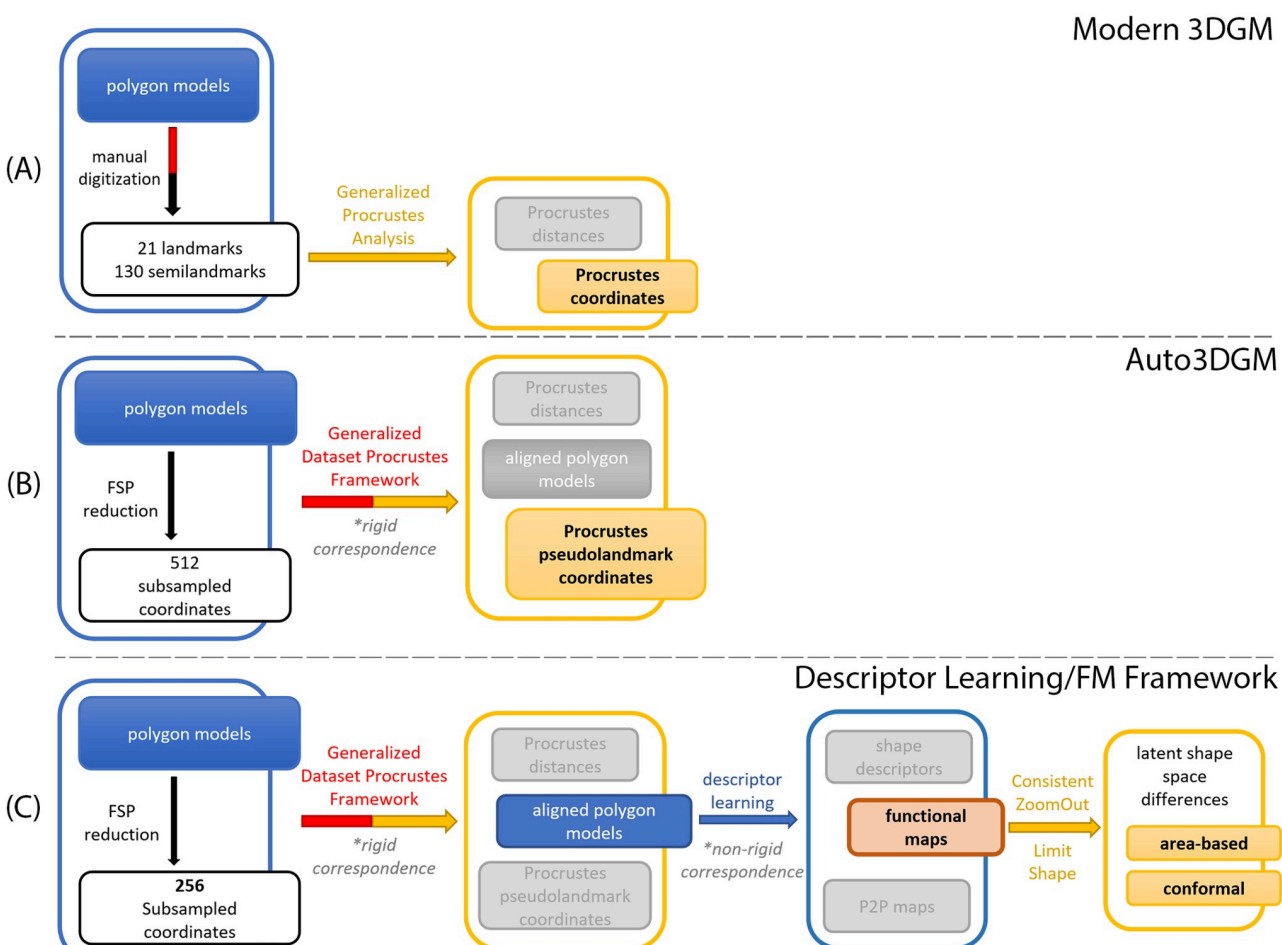

**Fig 1. Dataflow graphs of shape analysis pipelines used.** Rounded rectangular objects represent data objects and arrows depict the procedures/ algorithms operating on or producing them. All three analyses begin with triangular mesh models of bone. In **(A)**, triangular mesh models are manually digitized. The configurations of corresponding/homologous points obtained are subjected to Generalized Procrustes analysis (GPA) to yield Procrustes pseudolandmark coordinate shape variables. **(B)** represents automated quantification using Auto3DGM. Farthest point sampling (FSP) is used to subsample triangular meshes. The Generalized Dataset Procrustes Framework (GDPF) of auto3DGM assigns correspondences and aligns subsampled shapes to a common coordinate system (rigid alignment). **(C)** outlines the proposed descriptor learning approach that builds on auto3DGM with learned non-rigid/deformable functional correspondences between aligned polygon models. These functional maps are used to estimate latent shape space differences that characterize morphological variation expressed as area-based and conformal operators. Note that the GDPF step here subsamples shape at a low resolution with only 128 and 256 pseudolandmarks for initial and final alignment steps, respectively. While not sufficient for capturing shape variation, low-resolution auto3DGM produces the rotation and translation information needed to rigidly align our polygon models.

reduce the inherent complexity of morphology down to a limited set of phenotypic structures. Hence, they cannot describe all aspects of a bone's shape. Although expertly chosen and theoretically grounded given the use-case, landmark configurations assume that the researcher has *a priori* knowledge of what morphological features are important, which is often not the case [9, 12]. Gaining expert knowledge about phenotypic structures and then measuring them is often arduous, costly, and time-consuming. Notably, the landmarking process is prone to observer bias, and inter-and intra-observer error in identifying homologous landmark-based features can distort results [8, 9]. As our questions begin to warrant (i) a comprehensive, unbiased measurement of morphology, (ii) quantification of more complex skeletal elements or structures without *a priori* assumptions about feature importance, and (iii) an ability to

capture relevant shape variation for more extensive samples of complex biological shapes, researchers are increasingly looking to automated morphological phenotyping techniques as a solution [8, 13].

The demand for automated morphological phenotyping methods has lead to the publication of several viable solutions to the problem. Many of these new methods seek to detect landmark placement/position with little to no input from a trained morphologist [14, 15], while others attempt 'landmark-free' perspectives to morphological phenotyping [16–18]. These approaches may use some parameterization of a template or atlas specimen to inform the model, or, in a learning-based framework, supervising the automated process with a dataset of manually digitized training examples. Thus far, the only approach to morphological phenotyping that does not require a template or manual digitization, and attempts to quantify morphology comprehensively/exhaustively is auto3DGM [10]. This landmark-based algorithm represents each bone with a set of feature points uniformly and evenly sampled from each surface. It frames the task of aligning these 'pseudolandmarks' as an optimization problem. Automation hinges on the Iterative Closest Point-like alignment of configurations of pseudolandmarks that reasonably match corresponding points to each other [19]. auto3DGM substitutes modern 3DGM's sparse set of expert identified landmarks and semilandmarks for dense sets of algorithmically determined pseudolandmarks. These pseudolandmarks are aligned to yield Procrustes pseudolandmark coordinate shape variables much like those of modern 3DGM. While powerful, auto3DGM requires hundreds to thousands of pseudolandmark points to capture shape variation with adequate detail. This can be computationally costly and time consuming when sample sizes are large.

This study develops and validates a methodological first step towards automating morphological phenotyping and conducting morphometric analysis in the Functional Map (FM) Framework of Geometry Processing [20–22]. Ovsjanikov et al. 2012 present a procedure that relies upon functions defined on shapes, rather than points identified on shapes, to approximate their non-rigid or deformable correspondence. Correspondences between full triangular meshes or polygon models are expressed as linear operators between spaces of functions permitting the holistic study of differences in shape [21, 22]. In modern 3DGM, an expert morphologist identifies a sparse set of corresponding or homologous points on each polygon mesh and registers them via Generalized Procrustes analysis (GPA). An automated FM-based approach estimates correspondences between whole geometric objects algorithmically. Given correspondences, we can characterize the shape variation and tackle a host of statistical shape analysis tasks. FM-based shape analyses are now being used to capture and analyze variation in synthetic and real shape collections [23–28]. This study extends these methods to the analysis of collections of biological shapes.

Our primary contribution is a novel learning-based approach to producing highly accurate FM correspondences between biological shapes directly from their polygon model geometry. We then use functional map network (FMN) analyses to characterize shape variability [21]. In addition to expressing correspondences between shapes, FMs can be manipulated to express shape differences [29]. Shape difference operators are descriptions of shape deformations encoded as changes in two inner products between functions. These linear operators capture the disparity between shapes and decompose it into area-based shape differences (local changes in the area of the object's surface) and conformal (angular) shape differences (local changes in the angles between vectors tangent to the surface) [22]. The vectors of measurements whose inner products are the area-based and conformal shape differences, respectively, can be viewed as shape variables, comparable to Procrustes aligned coordinates, since they capture useful geometry and are invariant to isometries [21]. For simplicity, we call this shape analysis pipeline Morphological Variation Quantifier (morphVQ).

In this paper, we assess how suitable morphVQ and the functional mapping framework are to automated morphological phenotyping and to the study of complex shape variation. We compare the conformal and area-based shape variables obtained from morphVQ to shape variables obtained with manual digitization and with auto3DGM [10]. We assess the morphometric performance of our characterizations of shape by considering: (i) how accurate our approach is relative to auto3DGM as a benchmark, (ii) how the shape spaces formed by LSSDs compare to those of modern 3DGM and auto3DGM, (iii) how well we can predict *a priori* biological groupings from them, (iv) how between-group morphological differences obtained from them compare to those evidenced by modern 3DGM and auto3DGM, and (v) and how well our approach generalizes to morphological datasets with meshes that vary in size, shape complexity, and topological quality. With these criteria in mind, we find that our characterization of shape variation is both accurate and robust as it is comparable in morphometric performance to representations obtained via manual digitization and to auto3DGM.

## 2 Results

This study compares several morphometric analyses of the same 102 hominoid cuboids. The cuboid bone—often referred to as the hominoid foot's 'keystone' element—sits laterally in the primate midfoot. Biological anthropologists are interested in cuboid form because there is a robust evolutionary association between primate cuboid morphology, pedal joint function, and inferred locomotor behavior [30–39]. As such, researchers are interested in how cuboid shape and size covary with other morphological, functional, or behavioral phenotypes, as fossil cuboids can provide insights into the locomotor behaviors of extinct taxa. Automatically and exhaustively quantifying hominoid cuboid shape is a worthwhile real-world task for developing and evaluating our morphological phenotyping approach.

To establish a baseline for comparison, we capture two representations of hominoid cuboid shape using the expert identified landmarks and semilandmarks of modern 3DGM (see Methods and materials for details) (Fig 1A). The first uses 21 type I-III landmarks placed on discrete features on each polygon mesh. The second captures proximal and distal cuboid facet shape using 130 sliding surface semilandmarks. These landmark configurations have been used in previous studies to quantify cuboid shape variation [40]. We consider them ground-truth representations for our series of comparisons because they are based on specific cuboid feature importance models.

### 2.1 auto3DGM and rigid alignment

auto3DGM plays two roles in this study. In its first role, auto3DGM generates Procrustes pseudolandmark coordinates based on dense subsamples of points from the surface of each shape in our collection (Fig 1B). These algorithmically obtained shape variables directly characterize shape variation, so they are retained for our comparisons. With auto3DGM, we find that a sufficiently adequate analysis of hominoid cuboid shape requires at least 512 pseudolandmarks to quantify surface shape well. With greater than 512 points and a moderately sized collection of shapes, this procedure can become computationally prohibitive.

In its second role, auto3DGM is used to rigidly align our collection of polygon models to a global coordinate system. Our algorithm performs best when polygon models are approximately rigidly aligned as this avoids extrinsic symmetries that can't be disambiguated by our learning step [41]. auto3DGM's Generalized Dataset Procrustes Framework (GDPF) propagates rotation and translation information obtained during the alignment of pseudolandmarks to the polygon models themselves. We find that auto3DGM can produce useful rigid alignments between our hominoid polygon models in our collection with as few as 256

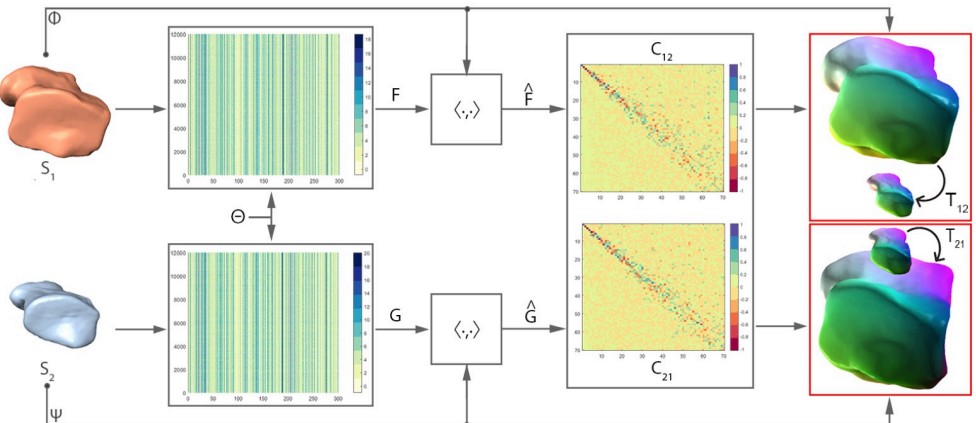

**Fig 2. Deep Functional Maps network architecture demonstrating functional and soft P2P map estimation in both directions.** We start with an initial pair of source and target shapes $S_1$ and $S_2$, respectively. $\Theta$ is a Siamese harmonic surface network, and $\Phi$ and $\Psi$ are the truncated Laplacian eigenbases for $S_1$ and $S_2$. Learned spatial descriptors are then projected to their corresponding bases to form $F$ and $G$. $C_{12}$ and $C_{21}$ are 70x70 functional maps (FMs) estimated in the forward and backward directions between source and target. On the far right are the recovered P2P maps $T_{12}$ and $T_{21}$, respectively. In P2P maps, visual representation of correspondence is demonstrated between (homologous) features that have the same color.

pseudolandmarks. While the Procrustes pseudolandmark shape variables from a low-resolution analysis don't characterize shape with enough surface detail, the aligned polygon models produced by the procedure are used as input to the descriptor learning step (Fig 1C).

## 2.2 Descriptor learning

Our descriptor learning model (Fig 2), is a variant of SURFMNet (see Methods and materials for additional details). SURFMNet is a Siamese neural network that learns to improve upon precomputed spectral shape descriptors to produce more accurate functional correspondences between shapes [42]. Our model differs from SURFMNet in two ways: (i) it accepts geometric data derived from rigidly-aligned polygon models as input as opposed to precalculated spectral shape descriptors, and (ii) it uses a Harmonic Surface Network (HSN) as a feature extractor in place of a fully connected residual network. We choose HSN as our feature extractor due to its ability to produce rotation-invariant features from polygon model geometry, an essential property for our descriptors (see Methods and materials for additional details) [43].

Our network's FM layer estimates the forward and backward correspondences, $C_{12}$ and $C_{21}$, between a source shape $S_1$ and a target shape $S_2$. These functional maps are easily converted to dense P2P correspondences, $T_{12}$ and $T_{21}$. Our approach to descriptor learning is practical, with learned descriptors producing consistent functional maps for all pairs of hominoid cuboid shapes in our collection.

## 2.3 Improving correspondences with Consistent ZoomOut

The P2P maps ($T_{12}$ and $T_{21}$) obtained with the learned descriptors contain artifacts. These artifacts are visible in the correspondences between pairs of shapes presented in Fig 3A.1 and 3A.2. Published studies using FM in computer graphics usually minimize such artifacts with some sort of post-processing refinement step [25, 44, 45]. We adopt the Consistent ZoomOut refinement technique, which iteratively and interchangeably optimizes the P2P and functional maps at multiple scales given the initial functional maps $C_{12}$ and $C_{21}$. Compared to their

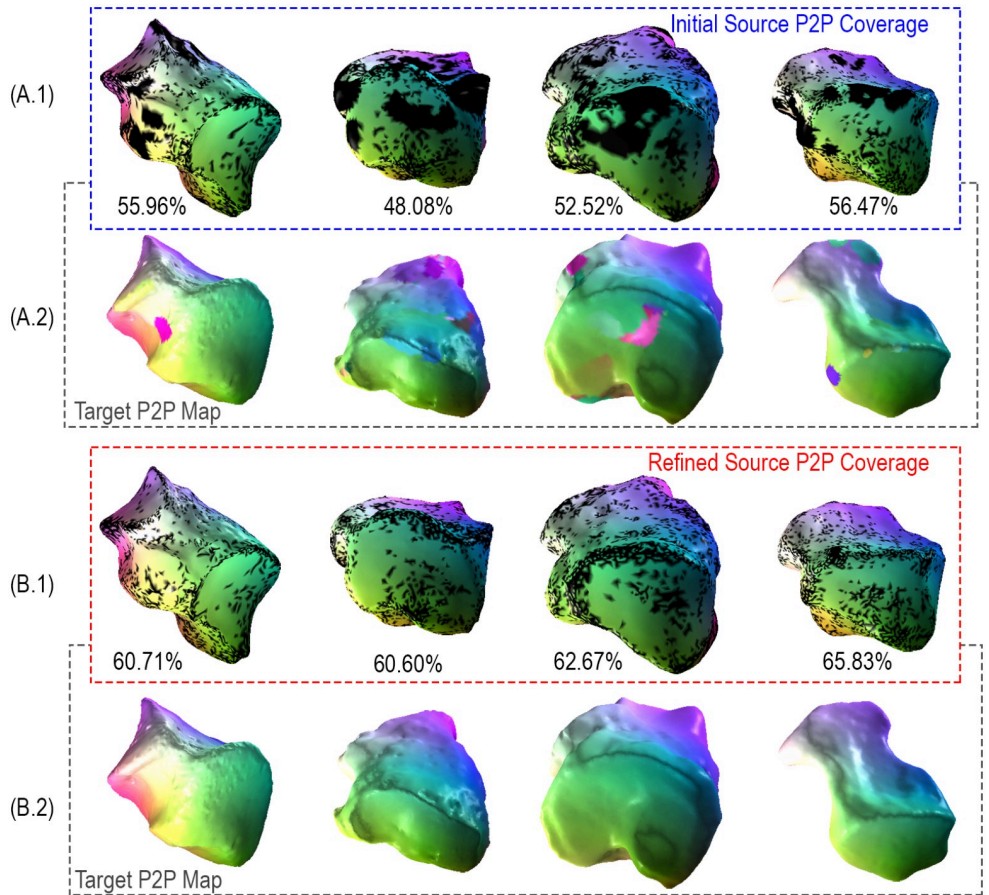

**Fig 3. Improving correspondence with Consistent ZoomOut.** We compare maps generated by HSN ResUNet descriptor learning model to their Consistent ZoomOut-refined counterparts for four randomly chosen pairs of hominoid cuboids. Rows **3A.1** and **3A.2** show source and target shape pairs produced by our model, respectively. Rows **3B.1** and **3B.2** show the same source and target shape pairs after Consistent ZoomOut refinement. Between shape pairs, surface regions of the same color (green/purple/pink/yellow) are considered homologous. Black areas on source shapes indicate a lack of bijective coverage with its associated location on the corresponding target.

counterparts in Fig 3A.2, the Consistent ZoomOut refined P2P maps in Fig 3B.2 achieve a smoother representation with fewer artifacts. Furthermore, we observe that these smoother representations are associated with increased bijective coverage as there is a 4% to 12% improvement when Consistent ZoomOut refinement is applied. Here bijective coverage refers to the ratio of the number of points in a source shape with a corresponding unique point on the target shape to the total number of points on that source shape. The Consistent ZoomOut refinement procedure hinges on a Limit Shape functional map network (FMN) analysis. Limit Shape characterizes shape variation to yield our area-based and conformal latent shape space differences (LSSDs) (see Methods and materials section for more information) [26, 29, 45].

## 2.4 Robustness to topological differences and mesh quality

Specimens within collections of real bone polygon models are often disparately sourced and can vary considerably in shape, size, resolution, and topological quality. These polygon model properties have been known to influence correspondence quality in other functional map-based methods [46]. Like most other methods, we alleviate correspondence estimation issues

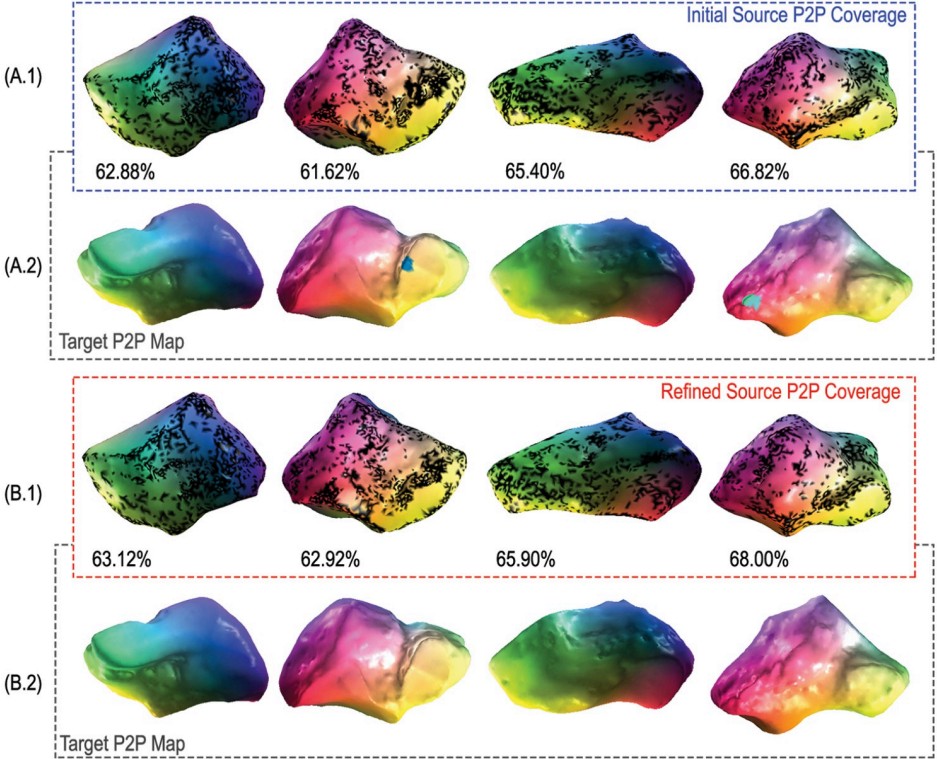

**Fig 4. Estimating and improving hominoid medial cuneiform correspondences.** We compare maps generated by HSN ResUNet descriptor learning model to their Consistent ZoomOut-refined counterparts for four randomly chosen pairs of hominoid cuboids.

that may be due to size differences by standardizing each mesh to unit surface area. To demonstrate our method's robustness to topological complexity and variability we apply morphVQ to two additional datasets. The first is a collection of hominoid medial cuneiforms that vary considerably in surface quality and resolution. As shown in Fig 4A.1 (source cuneiforms) and Fig 4A.2 (target cuneiforms), we achieve strong correspondence quality even with such irregular surfaces. While the refined correspondences of Fig 4B.1 and 4B.2 show only a marginal improvement in coverage, surface artifacts are consistently removed by the Consistent ZoomOut refinement process. The second dataset contains high resolution surface meshes of mouse humeri obtained from micro-CT scans. The relatively featureless humeral shafts present a correspondence challenge for functional map-based methods. Fig 5A.1 (source humeri) and Fig 5A.2 (target humeri) show high correspondence coverage, even along the relatively featureless shaft of the bone.

## 2.5 Shape space comparisons

In Fig 6, we visually compare shape spaces derived from three modes of hominoid cuboid shape quantification with individuals colored by genus. All six plots show distinctive groupings among hominoids. *Hylobates* and *Homo* separate well from all other groups in all shape spaces while *Pan*, *Gorilla*, and *Pongo* overlap to varying degrees. We use Pearson's r to assess the correlation between the manually digitized 21-landmark representation (Fig 6A)—a proxy for a ground-truth measurement—and our other representations from sliding semilandmarks, auto3DGM, and LSSDs (Fig 6B–6F). As expected, the highest correlation exists between the 21

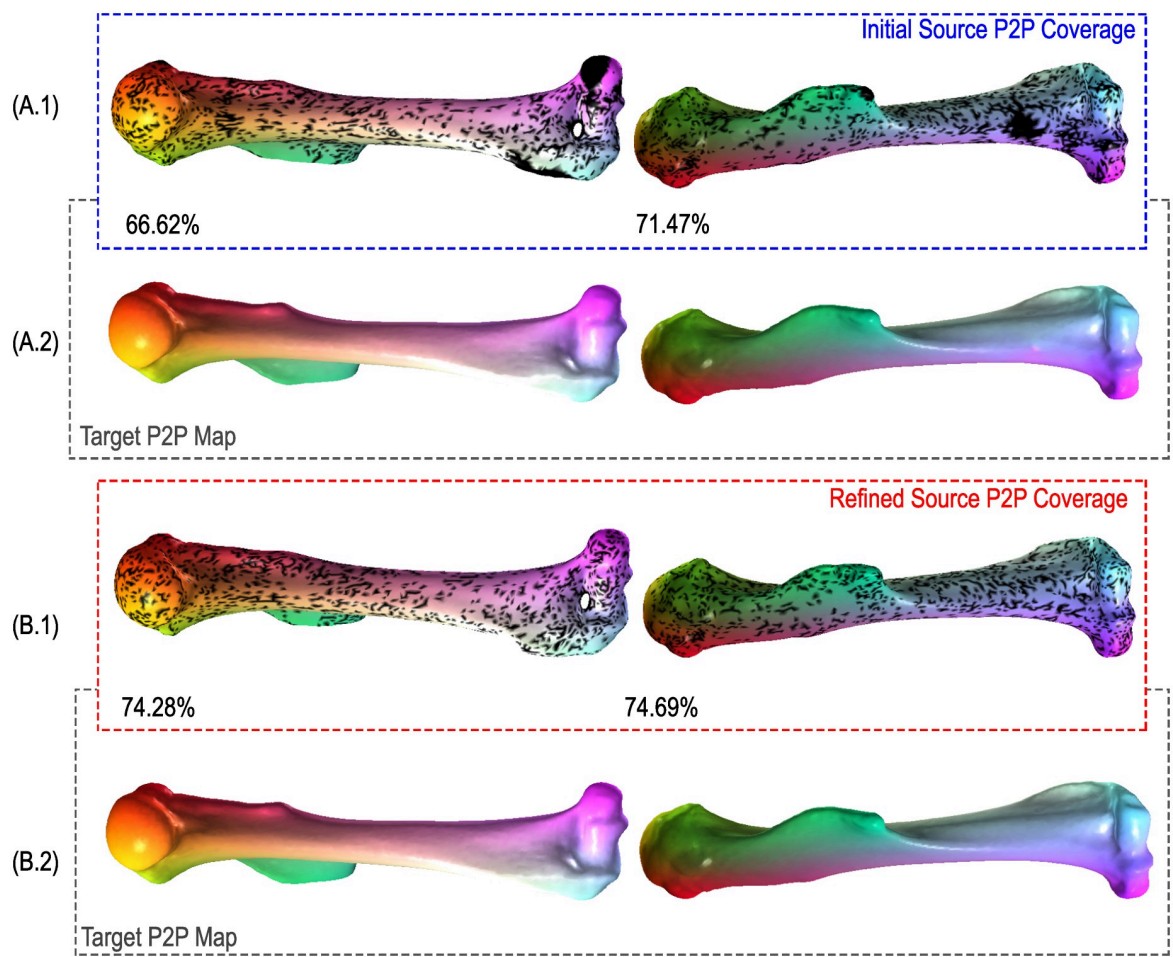

**Fig 5. Estimating and improving mouse humeri correspondences.** We compare maps generated by HSN ResUNet descriptor learning model to their Consistent ZoomOut-refined counterparts for two randomly chosen pairs of mouse humeri.

manually-placed landmarks Fig 6A and the other manually digitized sliding semilandmarks Fig 6B, $r(100) = 0.639$, $p < 0.001$. Of the representations derived from automated methods, the conformal LSSDs are most correlated with the 21-landmark representation, $r(100) = 0.621$, $p < 0.001$. The auto3DGM 512-pseudolandmark representation Fig 6D and the area-based LSSDs Fig 6E are only slightly less correlated at $r(100) = 0.616$, and, $r(100) = 0.618$, $p < 0.001$, respectively. We also assessed the proportion of variance each hominoid group accounts for in each representation using the trace of each group's variance-covariance matrix as a measure of spread. We found no difference in the variance partitioning between representations, i.e., groups account for the same proportion of total variance in each set of shape variables.

## 2.6 Classifying known biological groupings

The resulting PC projections are then classified by genus using four standard machine learning algorithms: K-means, Logistic Regression, Naive Bayes, and Linear Discriminant Analysis (LDA) (see Methods and materials section for more information). The resulting accuracies (average and standard deviation across eleven folds (Table 1). All representations perform well at genus classification. As expected, the first representation based on manually digitized

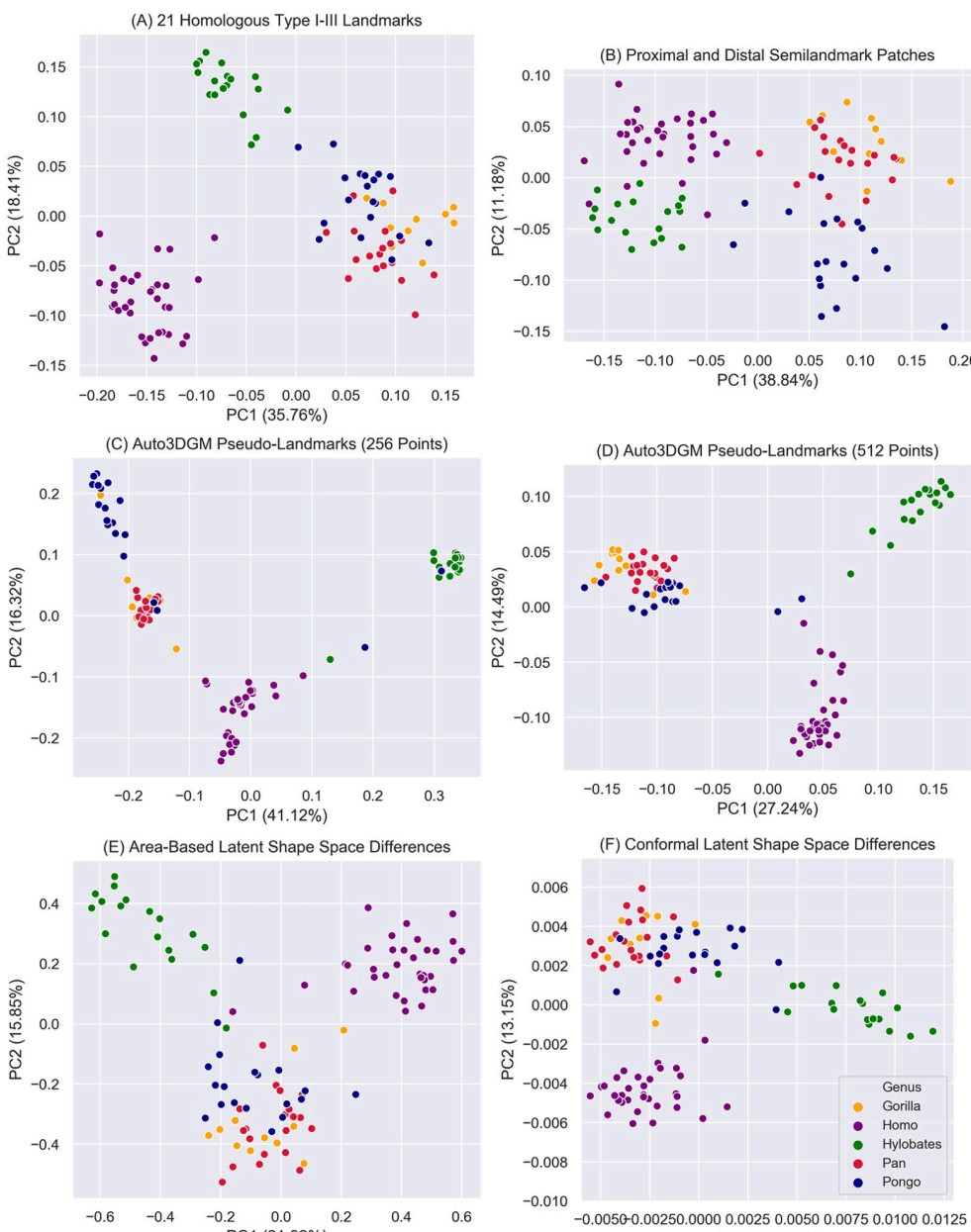

**Fig 6. Shape space differences between manually digitized and automated morphological quantification approaches.** The principal component (PC) scores PC1 and PC2 plotted in **A** and **B** are from the Procrustes aligned coordinates of the manually digitized landmarks. **C** and **D** are based on Procrustes-aligned pseudolandmarks obtained from auto3DGM analyses with 256 and 512 points, respectively. **E** and **F** are PCs obtained from our LSSD characterization. **E** is a morphospace derived from area-based differences in hominoid cuboid shape, while **F** is based on conformal shape differences.

landmarks performs best with each classification algorithm and our traditional 130-semiland-marks representation of cuboid shape yields the high accuracies across classifiers as well.

The automated representations—Auto3DGM at 256- and 512- pseudolandmark resolutions, Area-based LSSDs, and Conformal LSSDs, and both sets of LSSDs combined—all perform well at genus classification. Our Conformal LSSDs boast the highest classification

**Table 1. Accuracies and standard deviations for hominoid group classification task.**

| Quantification Mode | Shape Representation | K-Means* | Logistic Regression | | Naïve Bayes | | LDA[1] | |
|---|---|---|---|---|---|---|---|---|
| | | Accuracy | Accuracy | Std. | Accuracy | Std. | Accuracy | Std. |
| Manual digitization | Homologous Landmarks (21 points) | 0.99 | 0.981 | 0.044 | 0.939 | 0.076 | 0.974 | 0.052 |
| | Semilandmark Patches (130 points) | 0.932 | 0.993 | 0.026 | 0.976 | 0.049 | 0.992 | 0.028 |
| Automated measurement | Auto3DGM Pseudo-Landmarks (256 points) | 0.627 | 0.975 | 0.05 | 0.952 | 0.069 | 0.924 | 0.093 |
| | Auto3DGM Pseudo-Landmarks (512 points) | 0.906 | 0.972 | 0.054 | 0.922 | 0.084 | 0.86 | 0.118 |
| | Area-Based LSSDs (70x70 functional map) | 0.632 | 0.974 | 0.051 | 0.896 | 0.096 | 0.821 | 0.128 |
| | Conformal LSSDs (70x70 functional map) | 0.931 | 0.982 | 0.044 | 0.915 | 0.088 | 0.83 | 0.128 |
| | Area-Based & Conformal Combined[2] | 0.93 | 0.979 | 0.047 | 0.897 | 0.094 | 0.827 | 0.128 |

PCs corresponding to 95% of shape variation are used for each classification experiment (see Methods and materials for details).

[1]Linear Discriminant Analysis, [2]Area-based and Conformal LSSD features are combined and scaled.

accuracy (98.2%) on the logistic regression classification task. Notably, there is no significant change in classification performance when the LSSD shape variables are combined.

## 2.7 Highlighting where variability happens

We used the conformal and area-based LSSDs obtained to explore hominoid cuboid shape variability. Given our collection of shapes, the FMs obtained by our approach, an arbitrary FMN, and the LSSD shape variables, we can obtain shape variability across pairs of hominoid subgroups expressed as weighted distinctive functions projected directly on the bone surface (see Methods and materials for details) [26]. These distinctive functions highlight the locations where intrinsic distortions deviate most from the average distortion between hominoid groups.

First, for visual reference, we generated polygon models of group-mean shape configurations for each GM-based approach (Fig 7). Using our approach, we can then show distinctive functions between groups highlighting where variability happens (Figs 8 and 9). We compared the relative locations of the most distinctive between-group differences detected from modern 3DGM and auto3DGM to those areas identified by our function-based analysis (summarized in Table 2). Overall, we found that the surface regions highlighted by our FM-based shape analysis are the same cuboid surface regions that are most different between hominoid groups in modern 3DGM and auto3DGM analyses. For instance, the primary shape difference detected between *Pan* and *Homo* in both analyses occurs at the proximal facet's plantar-beak and along the distal portion of the medial aspect.

## 2.8 Validation via estimated points to ground-truth landmarks

As described in section 4.9, we obtain the means and standard deviations of the estimations of the five expert-placed landmark ground-truths generated by MorphVQ and Auto3DGM and summarize them in Table 3, as well as providing the descriptions of the markers for those values (Fig 10). Additionally, we include a box-plot (see Fig 11) to visually compare the difference between the medians (and the first and the third quantiles) of the estimations.

In summary, we find that MorphVQ provides consistent smaller mean estimations on 4 out of 5 markers as compared to the ground-truth, and also provides smaller standard deviations (see Table 3). The box-plot (Fig 11), on the other hand, indicates the estimations from MorphVQ generates smaller medians and quantiles than the Auto3DGM's estimations.

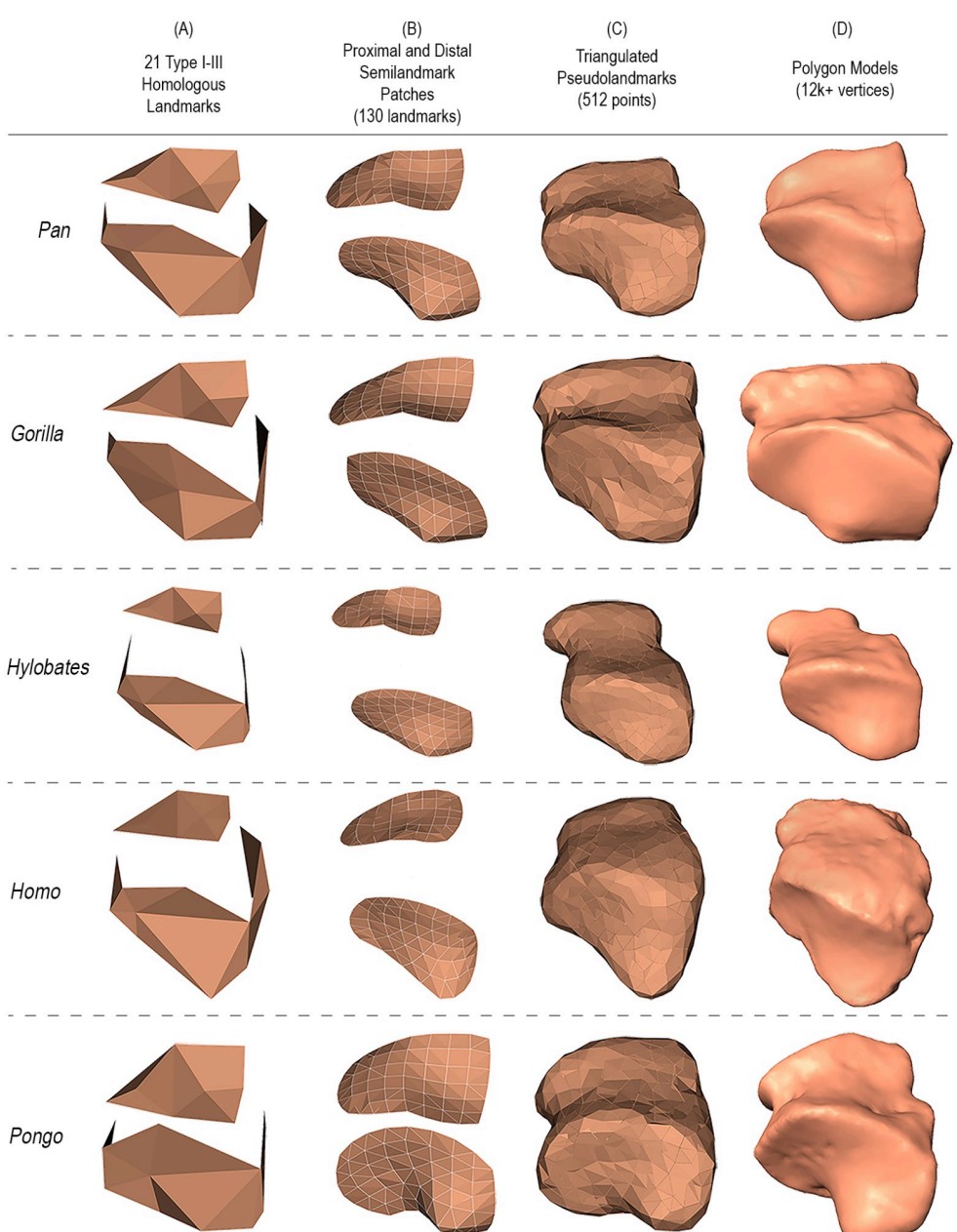

**Fig 7. Polygon models of hominoid cuboid group-mean shape representations. A** and **B** are polygon models based on landmark and semi-landmark patches where the vertices are manually digitized points. **C** shows pseudolandmark points (the vertices) with Delaunay triangulation. **D** features a randomly selected cuboid polygon model from each group for reference in the same orientation.

## 2.9 Basic speed comparison

While our method contains multiple steps, we are able to adequately quantify the cuboid dataset in less time compared to automated capture using Auto3DGM. Given our dataset of 105 cuboid meshes, it took 31.66 days to successfully use Auto3DGM to quantify shape variation using 512 pseudolandmarks. The Auto3DGM rigid alignment step at the beginning of our pipeline uses 256 pseudolandmarks and takes just 53.45 minutes to complete. The descriptor

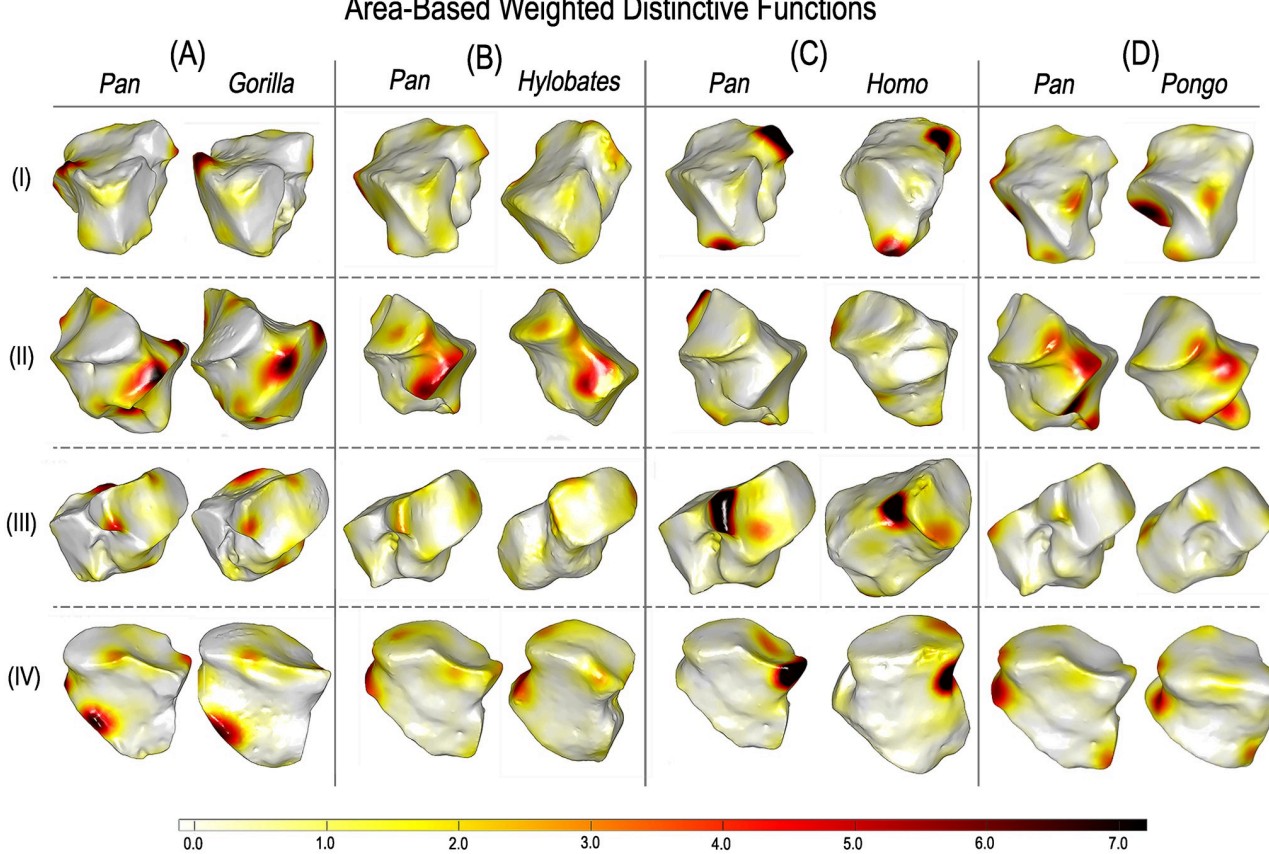

**Fig 8. Weighted distinctive functions highlighting where regions are most variable in area-based distortion between hominoid groups.** Dark red indicates the most variability, white the least. Rows: I Proximo-medial view; II Lateral; III Medio-distal; IV Dorso-distal.

learning step that follows rigid alignment takes approximately 1 day to complete, and the Consistent Zoomout/Limit Shape procedure at the end of the pipeline takes approximate 16 hours to yield our Area-based and Conformal LSSDs.

## 3 Discussion

This work demonstrates that it is feasible to automate morphological quantification in the FM framework of geometry processing. We used our descriptor learning model to produce high-quality spectral shape descriptors and FM correspondences between our hominoid cuboid polygon models. With these correspondences, we characterized shape variation within the shape collection using Limit Shape-based statistical shape analysis [26]. We found that the conformal and area-based LSSD shape variables perform as well as, or better than those obtained from 3DGM and auto3DGM. Therefore, we demonstrate an efficient, automated solution to capturing shape variation. LSSD shape variables capture the common landscape populated by the collection of polygon models, and there is a well-defined notion of distance between polygon models in this shape space [21, 22].

An FM-based shape analysis pipeline allows us to automate and standardize morphological phenotyping without manually digitizing each bone. Expert observation, interpretation, and digitizing are no longer rate-limiting steps in morphometric analyses [10]. With functional correspondences estimated between whole triangular meshes, we can analyze shape variation

## Conformal Weighted Distinctive Functions

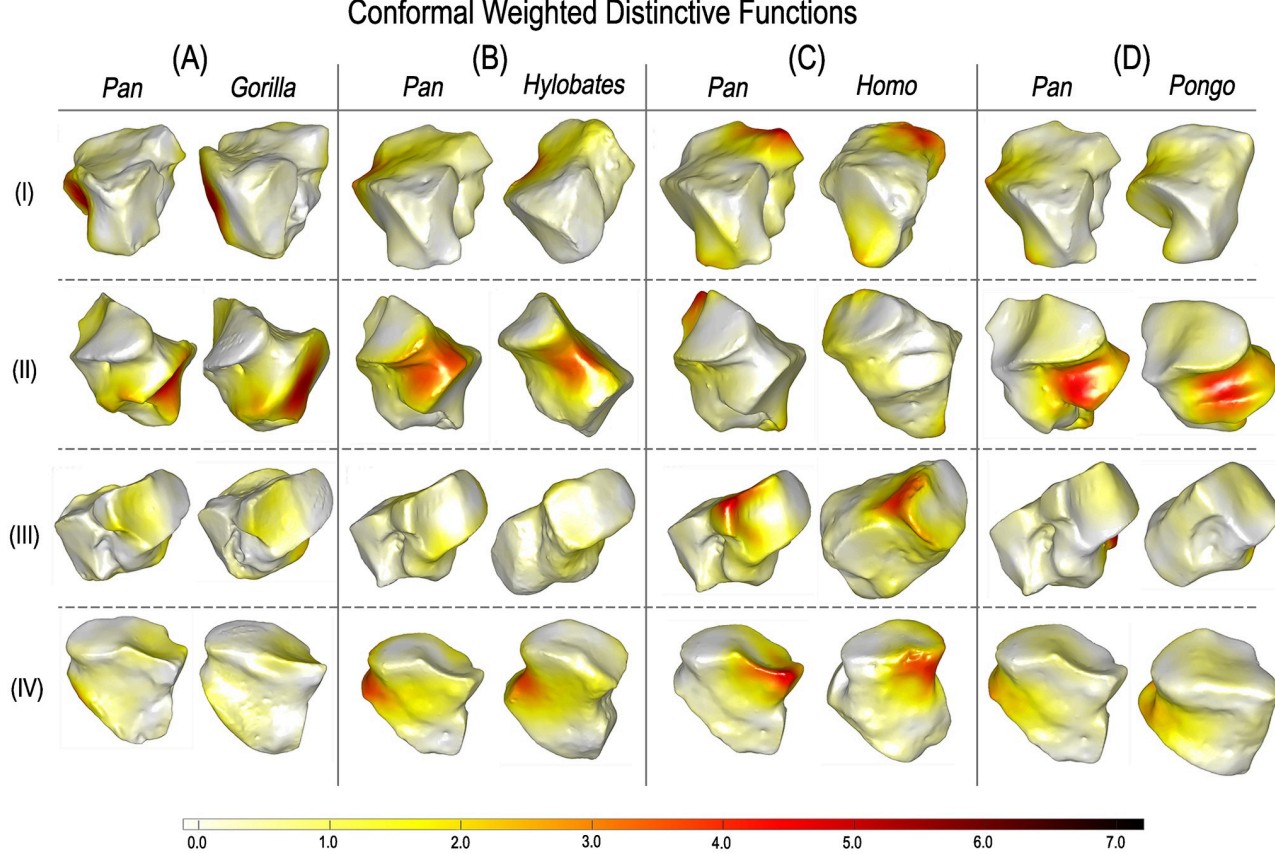

**Fig 9. Weighted distinctive functions highlighting where regions are most variable in conformal distortion on randomly selected pairs of hominoid cuboids.** Dark red indicates the most variability in surface angulation, white the least. Rows: I Proximo-medial view; II Lateral; III Medio-distal; IV Dorso-distal.

comprehensively and exhaustively since we are not limited by the practical and representational limitations of a reduced set of digitized landmarks. We are now permitted to ask questions and test hypotheses about the structuring of morphological variation based on robust evidence with fewer assumptions about which shape features are essential to sample.

The similarities between our LSSD characterization and those representations based on manually-digitized landmarks and auto3DGM tell us that our method captures meaningful morphometric information. Based on visual clustering patterns, the shape spaces associated with our area-based and conformal LSSDs in Fig 4E and 4F, respectively, bear striking resemblances to the ones formed by manually digitized landmarks in Fig 4A and 4B. The between-specimen euclidean distances of our LSSD representations are highly correlated with landmark-based distances ($r = 0.621$), indicating that shape variation is structured similarly between these methods. This Pearson's r is high considering the fact that area-based and conformal LSSDs hold different information from landmarks and pseudolandmarks. Landmarks and pseudolandmarks contain less morphological detail than area-based and conformal LSSDs. The Area-based and conformal LSSDs decompose different aspects of shape, therefore neither is expected to be highly correlated with landmarks. Also, the area-based and Conformal shape differences predict specimen genus affiliation with accuracies of comparable magnitude to landmark-based representations, telling us that our LSSDs encode between-group

**Table 2. Prominent between-group shape differences detected in representations from each analysis.**

| Difference Type | Representation | *Pan-Gorilla* | *Pan-Hylobates* | *Pan-Homo* | *Pan-Pongo* |
|---|---|---|---|---|---|
| Group Mean Shape Difference | Type I-III homologous landmarks | 1. Reduction in plantar beak size 2. Broadening of MT4 and MT5 facets | 1. Proximodistal elongation of the lateral side | 1.Proximo-distal elongation 2. Size and orientation of facets 3. Plantar beak position | 1. Negligible differences detected. |
| | Proximal and distal semilandmark patches | 1. Negligible differences detected. | 1. Proximodistal elongation of the lateral side 2. Size and orientation of facets | 1.Proximo-distal elongation 2. Size and orientation of facets 3, Plantar beak position | 1.Plantar/lateral section of proximal facet 2. Relative rotation of proximal and distal facets |
| | auto3DGM pseudolandmarks | 1. Deepening of peroneus longus groove on the lateral side | 1. Proximodistal elongation of the lateral side 2. Mediolateral narrowing 3. Angular changes between proximal and distal facets | 1.Proximo-distal elongation 2. Medial edge of 4th MT facet. 3. Plantar beak of proximal facet | Negligible differences detected. |
| Variability localization* | Conformal LSSDs | 1. Lateral part of the proximal facet | 1. Lateral aspect of bone (including *peroneus longus groove*) | 1. Medial and dorsal edge of 4th MT facet 2. The plantar beak of the proximal facet | 1.Plantar/lateral section of proximal facet 2. Lateral aspect of bone (including *peroneus longus groove*) |
| | Area-based LSSDs | 1. Dorso-lateral margins of the proximal facet | 1. Lateral aspect of bone (including *peroneus longus groove*) | 1. Plantar beak of proximal facet 2. The medial edge of the 4th MT facet | 1. *Peroneus longus groove* |

They are enumerated (1, 2, etc.) in order of prominence. The first three shape rows describe landmark-based differences in group mean shape. The last two rows describe the locations of significant shape variability between groups as indicated by weighted highlighted functions (see Methods and materials for additional details)

differences just as well as manually-digitized measures. These similarities in variance structure and morphometric performance strongly suggest that our LSSD shape variables characterize the same biologically meaningful geometry captured by our specialized configuration of manually placed landmarks.

Morphologists find GM tools and methods appealing because they allow us to study shape differences and variability in ways that preserve geometry throughout an analysis [4, 5, 8]. With modern 3DGM, we can visualize the morphological variation we capture as landmark displacements in a low-dimensional embedding of shape space. By preserving intrinsic geometry, our FM-based approach affords the same biological utility. Albeit differently, conformal and area-based LSSDs encode detailed information about the disparity or distortion between shapes under their estimated correspondence. With the distinctive functions derived from LSSDs, we can explore shape differences by visualizing where variability localizes between groups. Instead of deforming an anatomical model of bone or some mean-configuration of landmarks to illustrate shape differences in a morphospace, distinctive functions highlight the modes and regions of shape variation directly on polygon model surfaces. This approach differs from other shape analysis methods because it recognizes morphological regions by patterns of variability rather than mean difference. Thus, this approach has a more direct application to evolutionary studies that seek to quantify morphological variability to test evolutionary hypotheses.

In comparing different species or groups of specimens, it is common to investigate which aspects of shape allow discrimination. With distinctive functions, we can identify the surface

**Table 3. Mean Euclidean distance between estimated points and ground-truth landmarks.**

| Landmark Description | MorphVQ Error | | Auto3DGM Error | |
|---|---|---|---|---|
| | Mean | Std | Mean | Std |
| 1. Most proximal point of plantar "beak" | 0.0731 | 0.0161 | **0.0715** | 0.0244 |
| 2. Center of proximal facet | **0.0607** | 0.0157 | 0.0674 | 0.0173 |
| 3. The most medial point of indentation between distal and proximal facets | **0.0512** | 0.0184 | 0.0749 | 0.0220 |
| 4. Deepest point of indentation on the facet for 5th metatarsal | **0.0381** | 0.0110 | 0.0692 | 0.0163 |
| 5. Deepest point of indentation on the facet for 4th metatarsal | **0.0607** | 0.0108 | 0.0717 | 0.0192 |

Landmarks 1–5 are placed on 69 sample cuboids and the means and standard deviations of the distances between those landmarks and their automated counterparts in MorphVQ and Auto3DGM are presented. The smaller of the two distances, emphasized in bold, identifies the automated approach with less error per landmark.)

regions where shape is most variable between groups. In particular, we can show the locations where area-based and conformal distortions deviate most from the average between-group distortion (Fig 4E and 4F (i.e., 'where variability happens' on the surface [23]). We found that our distinctive functions highlight the same hominoid cuboid regions where our modern 3DGM and auto3DGM analyses detect significant between-group differences in shape, providing further evidence that the algorithm is detecting biologically meaningful shape information (Table 2). With distinctive functions highlighting relevant variability, these methods permit automated detection of the regions or features that are morphologically distinctive for intergroup comparisons.

More persuasive evidence of the validity of our approach can be found in Fig 11 and Table 3 where we compare landmark estimation performance between the proposed MorphVQ and Auto3DGM. Here, we use the amount of error Auto3DGM introduces when pseudolandmarks are used to predict landmark positions as a baseline for high quality automated biological correspondence. MorphVQ obtains closer or even smaller mean/median errors of the landmark estimation relative to Auto3DGM. This demonstrates that our approach is capable of yielding landmark position estimates that are as good as those obtained from Auto3DGM where biological correspondence and homology are concerned. Furthermore, the smaller quantile ranges (Fig 11) and the smaller standard deviations (Table 3) of the estimation errors generated by MorphVQ indicate this model can generate more stable and concentrated inferences when compared to Auto3DGM.

We also find that learning spectral descriptor using an HSN feature extractor leads to highly bijective FM correspondences with good coverage between all pairs of polygon models. This is in comparison to solving directly for correspondence using precomputed Wave Kernel Signature (WKS) descriptor or to learning a better descriptor using FMNet or SURFMNet. Our results show that our method generalizes well across different datasets, even in situations where topological complexity, noise, and other surface related issues may cause the previously mentioned approaches to fail. This is because correspondence quality depends heavily on the geometric properties of the pointwise shape descriptor used to solve the problem. The WKS is a popular choice because it is multiscale, isometry invariant, easy to compute, and contains all intrinsic information at each point [47, 48]. Despite its benefit, there are drawbacks to using WKS from real-world polygon model representations of bone (see S3 Fig). Though for different reasons, the well-documented sensitivity and specificity issues of WKS and HKS, respectively, can lead to poor correspondence quality, especially with a dataset of disparately obtained bone polygon models that differ drastically in shape and triangulation [49]. For

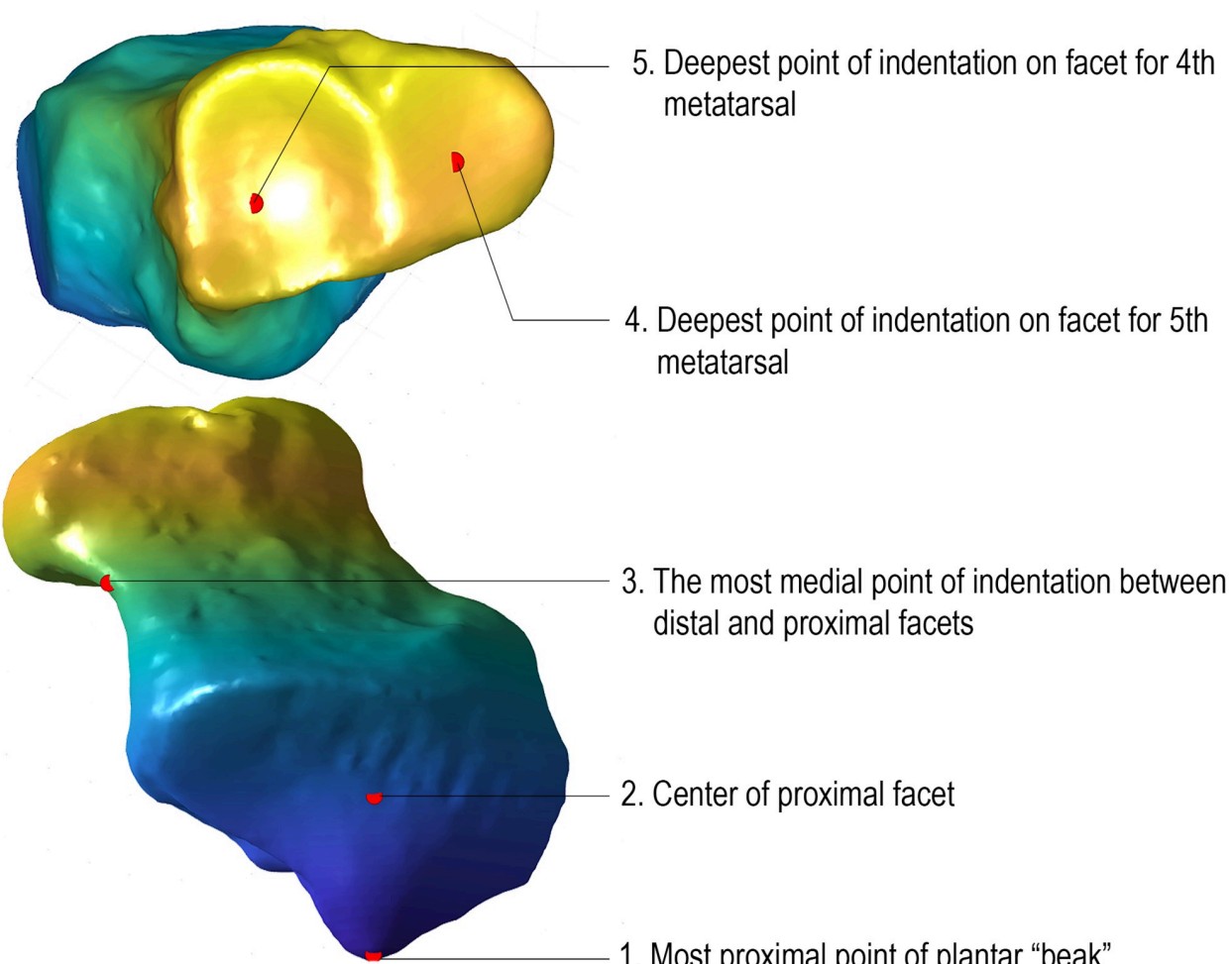

**Fig 10. Hylobates Cuboid showing validation landmarks.** Red dots indicate where landmarks are manually placed on 69 sample cuboid meshes.

instance, WKS descriptors yield high-quality bijective correspondences between similarly shaped *Pan* cuboids but produce poor correspondences between *Pan* and *Hylobates* cuboids as they are quite different in form. In contrast, our model encodes informative spectral descriptors that yield high-quality correspondences in both scenarios.

In summary, studies that characterize morphological phenotypes have relied on the analyses of manually digitized landmarks. Such analyses impose *a priori* constraints on which aspects of surface morphology can be captured, and an increasing body of evidence points to the fact that we need more than a few key traits to adequately characterize morphological variation [10, 12, 50–52]. We demonstrate that FM-based methods can automate comprehensive morphological quantification and provide a nuanced analysis of intrinsic shape variability. With efficient descriptor and correspondence learning, and FMN-based analysis tools like Limit Shape, we can make significant advances toward expanding the GM toolkit to include landmark-free analyses of biological shapes.

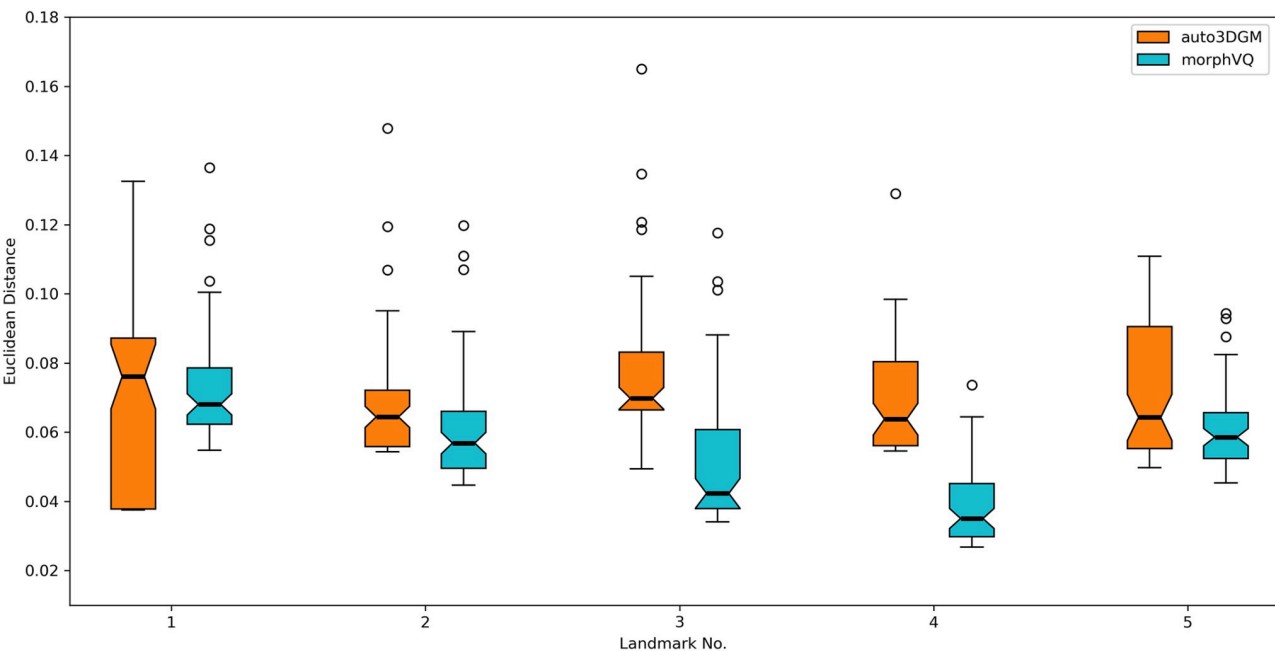

**Fig 11. Boxplot of Validation Results.** Red dots indicate where landmarks are manually placed on 69 sample cuboid meshes.

# 4 Materials and methods

## 4.1 Data acquisition and preprocessing

Our study sample consists of 102 triangular meshes obtained from laser surface scans of hominoid cuboid bones. These cuboids were from wild-collected individuals housed in the American Museum of Natural History, the National Museum of Natural History, the Harvard Museum of Comparative Biology, and the Field Museum. *Hylobates*, *Pongo*, *Gorilla*, *Pan*, and *Homo* are all well represented. Each triangular mesh is denoised, remeshed, and cleaned using the Geomagic Studio Wrap Software [53]. The resulting meshes vary in vertex-count/resolution from 2,000—390,000. Each mesh is then upsampled or decimated to an even 12,000 vertices using the recursive subdivisions process and quadric decimation algorithm implemented in VTK python, respectively [54–56].

The first of the two smaller datasets used to evaluate generalizability is comprised of 26 hominoid medial cuneiforms meshes isolated from laser surface scans obtained from the same museum collections listed above. The second dataset is made up of 33 mouse humeri meshes sourced from micro-CT data (34.5 μm resolution using a Skyscan 1172). These datasets were processed identically to the 102 hominoid cuboid meshes introduced above. See the Dryad data repository at [57] for all data associated with this study.

## 4.2 Expert measured and auto3DGM cuboid form quantification

We used Stratovan Checkpoint Software [58] to quantify shape variation in two modes. The first configuration is of 21 well established type I-III landmarks placed on prominent points and facet margins, the second configuration consists of two semi-landmark patches placed on the proximal and distal articular facets only. The semilandmark patches in the second mode were anchored using 8 homologous landmarks for a total of 130. These sets of landmark

configurations were subjected to a generalized Procrustes analysis with semi-landmark sliding in the Geomorph R package [3, 6, 59, 60]. The Procrustes-aligned coordinates from both were retained for further analysis.

For comparison, we used the auto3DGM R package to capture hominoid cuboid shape variation at two resolutions, with 256- and 512-pseudolandmarks, respectively [10, 19]. The lower resolution analysis was initialized with 150 subsampled points and the higher resolution was initialized with 256 points. The Procrustes-aligned pseudolandmarks obtained from these shape analyses are then subjected to a separate principal component analysis (PCA). The PC scores and eigenvalues from both are retained for further analysis. We also retain the aligned polygon models produced by the lower resolution analysis for our descriptor learning procedure.

### 4.3 Functional maps and descriptor learning

Using MATLAB, we rescaled each auto3DGM aligned polygon model to unit area. We discretized each model using the cotangent weighting scheme to yield stiffness and mass matrices that are then used to compute the Laplace-Beltrami Operator (LBO) via eigendecomposition [61]. We retained the LBO and the raw polygon model geometry (vertices and triangles) for descriptor learning in the FM-based framework in our proposed model.

We use the FM framework because there are several advantages to solving the correspondence problem in the functional domain, especially with the LBO as a basis. The LBO, a generalization of the Fourier analysis on Riemannian manifolds, is the eigenbasis of choice as it is invariant to isometric transformations, and it is well suited for continuous maps between geometric objects [20, 23]. With the LBO as a basis we can use FM-based tools to efficiently transfer functions from a source to a target shape to avoid manipulating pointwise correspondences. The FM-based correspondence problem is linear and easy to optimize compared to the non-convex correspondence problem faced when points are considered. Using a truncated LBO with as few as 70 eigenfunctions reduces the dimensionality of the problem significantly without much loss to correspondence quality [22].

Given bones as source and target shapes, denoted by $S_1$ and $S_2$, the framework proposes the following general steps to calculate the functional maps:

1. First, we obtain $k_1$ and $k_2$ number of basis functions (LBO eigenfunctions) on the source and target shapes, respectively. We then project sets of precomputed shape descriptors (e.g., pointwise features that encode the local and global geometric properties of each surface) onto their respective bases to yield coefficients denoted $A$ and $B$.

2. The functional map framework focuses on the following equation to solve for $C$, the map that preserves the point-to-point map correspondence between the shapes:

$$C_{\text{opt}} = \arg \min_C E_{\text{desc}}(C) + \alpha E_{\text{reg}}(C) \tag{1}$$

Here are the eigenvalue matrices of the two sets of basis functions:

$$E_{\text{desc}}(C) = \| CA - B \|^2 \tag{2}$$

$$E_{\text{reg}}(C) = \| C\Lambda_1 - \Lambda_2 C \|^2 \tag{3}$$

3. Once the functional map is obtained, standard implementations then recover the point-to-point map by nearest neighbor search. Standard implementations improve the quality of

the point-to-point map using post-processing refinement tools such as Bijective Continuous Iterative Closest Point (BCICP) or Kernel Matching [25, 44]. These post-processing steps are bypassed in our implementation, in favor of refinement using the Limit-Shape based Consistent Zoomout method (see Implementation for details).

While estimating FMs in this way is efficient and practical for whole polygon models, it can be quite sensitive to the type of shape descriptor used. Shape descriptors encode the relevant local and global geometric properties of each point of a shape to a vector in some single- of multi-dimensional feature space [62]. There are multiple types of point-based descriptors of shape that provide initial points for mapping shapes to each other in the FM framework. These task-specific descriptors have specific geometric properties depending on their use case [49]. For example, spectral shape descriptors, which are derived from the spectral decomposition of the Laplace Beltrami Operator associated with shapes, are invariant to isometric transformations and are potent descriptors of intrinsic geometry. Such spectral descriptors are the current state-of-the-art [47, 63, 64]. Several studies show that it is possible to learn spectral descriptors directly and that learned spectral descriptors perform better in practice [42, 49, 62, 65].

The best performing descriptor learning models, such as FMNet and SURFMNet, are based on Siamese neural networks that transform pairs of shape descriptors into new ones to improve functional correspondence [42, 66]. Despite being learning-based, they still adhere to the three step FM-pipeline previously described. With precomputed SHOT (Signature of Histograms of OrienTations) or WKS (Wave Kernel Signature) descriptors as input, these models use a deep residual neural network (ResNet) with shared weights to produce new features that are then projected into their respective eigenbases to create descriptors. FMNet and SURFM-Net then use these new descriptors to estimate improved functional correspondences between shapes.

## 4.4 Learning intrinsic features from surfaces

Instead of using precomputed SHOT or WKS descriptors as the functions in a functional maps framework, several recent methods focus on learning spectral descriptors via the functional characterization of the vertices/point clouds of polygon models [41, 46]. These approaches use specialized neural network architectures (e.g., PointNet++) or novel convolutional kernels (GCCN, MDGCN. KP-Conv, Metric-based Spherical Convolutions, etc. [67–70]) capable of exploiting the geometric features of point-clouds or triangular meshes. These descriptor learning models replace the Siamese ResNet architecture of FMNet with spatial feature extractors that have been implemented in a Siamese way [46]. Given the unstructured point clouds, they create new features that are invariant to translation, rotation, and point order. Just as in FMNet, these features are then projected into the eigenbasis to form the spectral descriptors needed to estimate functional correspondence. Point-cloud-based feature extraction approaches such as these yield higher quality correspondences compared to their precomputed descriptor-based counterparts.

Despite the advantages, extracting features of sufficient quality for functional characterization is difficult. Existing methods either (i) suffer from poor expressivity or (ii) are too sensitive to differences in polygon model connectivity, or (iii) they don't produce rotation-invariant features in a manner that is conducive to learning spectral descriptors. In this study, we craft our own shape descriptors directly from polygon mesh geometry using the deep U-ResNet (HSN) architecture used for shape classification and segmentation introduced in [43]. HSN is one of many charting-based approaches that generalizes the notion of convolutions to irregularly sampled manifolds and graphs. With charting-based methods we can apply a convolution filter on the irregular, non-euclidean grid of a whole surface in the same way we apply traditional

filters to whole images. In addition to the other charting-based models, HSN possesses several other advantages: 1. It does not rely on orientation definition on the coordinates; 2. It does not define operation on euclidean spaces, which makes it a natural fit for cases with manifolds. Namely, HSN proposes a spectral-harmonic-based operator for convolution on surfaces that take the input meshes as inputs. For a give point on a mesh, the convolution considers its geodesic neighboring points and convolve with them by the spectral harmonic operator after converting those points via a parallel transport function, a necessary step to map all points into the same space for convolution. Notice that although the input meshes for HSN are only the realizations of the surfaces, all operators are designed to directly gather information on surfaces. And given the natural of the spectral harmonic operator, the output from this operator possesses rotation-invariant ($M = 0$ in the HSN setup) and rotation-equivariant ($M = 1$ in the HSN setup) properties. The above module is the convolution building block for the HSN, a U-ResNet-based architecture. It takes the mesh as inputs in our case, and outputs multidimensional shape descriptors.

Overall, an HSN-based feature extractor produces highly expressive intrinsic features that are strongly locally-aligned. In practice, compared to PointNet++, an HSN feature extractor encodes rotation-invariant features in a way that is more forgiving of arbitrary differences in pose (SO3 rotations) between a source and a target polygon model. This is usually the case for real-world pairs of bone meshes which may be poorly aligned, if at all. In practice, we find that descriptors from HSN-based features consistently yield highly bijective FMs with detailed and accurate pointwise correspondences.

## 4.5 Implementation

**4.5.1 Deep HSN UResNet feature extractor.**   Our U-ResNet architecture is deeper with eight stacks of ResNet blocks and three levels of pooling and unpooling layers (see S1 Fig) ResNet blocks are unchanged, each containing two HSN convolutional layers and a residual connection. Like Wiersma et al., we configure the network with rotation order streams: $M = 0$ and $M = 1$, where the former enforces rotation invariance and the latter, equivariance. In the scope of machine learning, the terms rotation invariance and equivariance usually refer to two properties that describe the relative rotational correlations with the original inputs, where the former indicate a constant encoding feature regardless of the rotations on the inputs and the latter indicate the encoding features will have corresponding rotations as their corresponding inputs. By using the HSN structure that considers both rotation invariance and equivariance, we identify ±5% degree of rotational variations to the input data. For our model configuration, we use 16-, 24-, 32-, and 48- units at each scale of the deep U-ResNet. Pooling and unpooling are done in the same way via parallel transport, but with pooling ratios 1, 0.5, 0.25, and 0.1 starting with a radius of 0.1 that grows by a factor of 2 at each scale. The 48-unit output produced by our last ResNet block then passes through the unpooling and reverse convolution layers of the U-net. This results in a 16-dimensional vector at each node of the original triangular mesh, which is then transformed to a 300-dimensional vector by a densely connected layer with ReLu activation. This HSN feature extractor was implemented using code from Wiersma et al., 2020 which was retrieved from https://github.com/rubenwiersma/hsn.

**4.5.2 Functional map layer and unsupervised loss.**   The Siamese HSN feature extractor produces 300-dimensional spatial embeddings at each vertex in our source and target shapes. We projected these embedding into their respective LBO eigenbases to form spectral descriptors, which are then passed to a fully differentiable functional map layer that computes the functional maps $C_{12}$ and $C_{21}$ (in both directions) between the source and target shapes. After obtaining $C_{12}$ and $C_{21}$, we use the axiomatic SURFMNet loss defined in Roufosse et al. 2019 w.

r.t. the model weights to optimize for the best $C_{12}$ and $C_{21}$. This is done directly in the functional domain.

This unsupervised loss consists of 4 regularization terms (or penalties) that assess the structural properties of the functional maps obtained from the FM-layer [42]. The first penalty, E1, enforces bijectivity by requiring that $C_{12}$ and $C_{21}$ be inverses of each other, which ensures that the composition of the two maps is an identity map. The second penalty, E2, constrains orthogonality. This condition preserves the local area of the two input shapes in the P2P map when converting back from the two functional maps. The third, E3, is a well-known regularizer in the functional map pipeline because it enforces commutativity with the Laplacian [20, 71]. The fourth, E4, guarantees that the learned correspondences in the form of functional maps directly arise from the P2P map. Specifically, it means both functional maps are commutable w.r.t. the reduced bases of the descriptors of the corresponding source and target shapes. The FM layer and the unsupervised losses were implemented using a PyTorch Geometric [72] library retrieved from https://github.com/pvnieo/SURFMNet-pytorch.

**4.5.3 Training scheme.** Following Wiersma et al., 2020, we precomputed the logarithmic maps, weights, and multi-scale radius graphs needed for HSN feature extraction from each auto3DGM aligned polygon model. We also retain the point-cloud/vertices of each aligned model as input to our network. Each model was trained using the ADAM optimizer with a learning rate of $1 \times 10^{-3}$ in a data-parallelized manner [73]. Models are trained on random pairs of shapes drawn from the set of all pairwise comparisons without replacement. An epoch is complete when each pair of shapes in the collection has been seen once. This is repeated until convergence in a self-supervised manner with no validation set. At convergence, each pair of shapes is processed to produce a full set of shape descriptors, and a pair of functional maps from source to target shape and the reverse. These are all retained for further analysis. These models were trained with a batch size of one on a NVIDIA Tesla K80 GPU core.

## 4.6 Refinement via consistent ZoomOut and limit shape

Once the functional maps (e.g. $C_{12}$ and $C_{21}$) are obtained via the proposed deep learning model, Consistent ZoomOut is used to refine the corresponding point-to-point mappings with the information of the functional mappings between the shapes in a collection. In particular, the Consistent Zoomout is an optimization technique set upon the functional maps with the following objective,

$$\min_{\mathcal{G}} E(\mathcal{G}).$$

Where

$$E(\mathcal{G}) = \sum_{(i,j) \in \mathcal{G}} \sum_{k} \frac{1}{k} \| C_{ij}^k \, C_{ji}^k - I^k \|_F^2$$

$$= \sum_{(i,j) \in \mathcal{G}} \sum_{k} \frac{1}{k} \| (\Phi_j^k)^\dagger \Pi_{ji} \Phi_i^k \, Y_i^k (Y_j^k)^\dagger - I^k \|_F^2.$$

We use the same color to refer to the expansion of the functional maps (e.g. $C_{ij}$) from different perspectives, where the orange one encodes the point-to-point map correlation and the cyan one encodes shape consistency with the limit shape $S_0$ (refer to Fig 12 for details). Here $\mathcal{G}$ indicates the set of all functional maps for the given set of shape collections; $i$ and $j$ refer to the corresponding two shape indices in this collection; $k$ refers to the dimension of the eigenbases (e.g. $\Phi_j^k$ and $\Phi_i^k$) in the functional space; $I$ is the identity matrix of dimension $k$; $Y_i$ is the

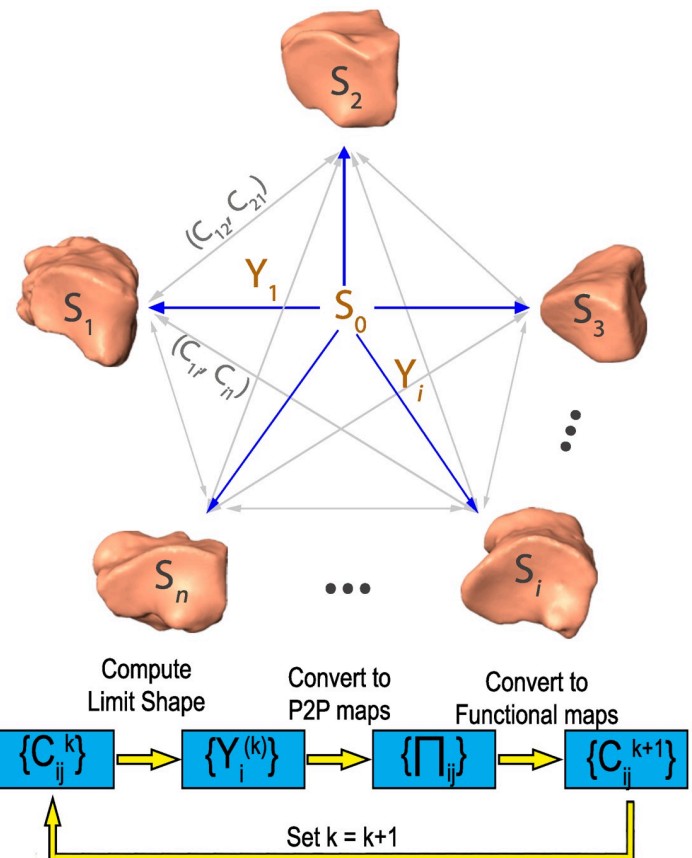

**Fig 12. Limit Shape and consistent ZoomOut Illustration.** Given a shape collection, Limit Shape $S_0$ is the "mean" shape considering all the shape variation within this shape collection via $Y_i$. $C_{ij}$ (or $C_{ji}$) is the functional map between shape $S_i$ and $S_j$. The bottom pipeline indicate the way Consistent ZoomOut is performed to refine the functional maps through a joint work of 1. Limit Shape Recomputing; 2. P2P map conversion given the new Limit Shape; 3. Convert back to functional maps from P2P map.

mapping between the shape $S_i$ to the limit shape $S_0$ with $C_{ij} \approx Y_j(Y_i)^\dagger$ for $\forall(i,j) \in \mathcal{G}$; $\Pi_{ji}$ is the point-to-point map between shape $S_i$ and $S_j$ and $\Pi_{ji}(p, q) = 1$ if and only if $T(p) = q, \forall p \in S_i, q \in S_j$. And $T(\cdot)$ is the projection function from points in $S_i$ to points in $S_j$.

Although Consistent Zoomout is formulated as a multi-scale optimization problem, in reality, the low quality (e.g. with smaller $k$) functional maps are the ones we have initially, which makes it infeasible to optimize the above equation directly. To solve this problem, the authors of the paper propose an optimization equivalence, which is an iterative functional map refinement procedure with the 3 major steps: 1. Form the Limit Shape matrices (e.g. $Y_i$) by considering all the functional maps (e.g. $C_{ij}^k$) at quality level $k$. 2. the Limit Shape matrices are then used for the generation of new point-to-point maps (e.g. $\Pi_{ij}$) within this shape collection. 3. The improved functional mappings (e.g. $C_{ij}^{(k+1)}$) are then generated from these new point-to-point maps. By repeating these 3 steps above, we are able to improve the quality of the functional maps and hence obtain refined point-to-point maps. Refer to (Fig 12) for diagram illustration of this process.

Notably, one key component of the Consistent ZoomOut is Limit Shape. As previously mentioned, it is used as an intermediate step to produce consistent point-to-point mappings in each of the iterations. The Limit Shape method's CCLB matrices [45] provide canonically

consistent bases across different shape inputs, normalizing for shape-dependent inconsistencies in the LBO. These techniques ensure the Consistent ZoomOut a superior algorithm over its predecessor, ZoomOut [28], which is biased w.r.t. the choice of source and target, and is designed only for pairwise analysis, ignoring the relationships between shapes in a collection. The Consistent ZoomOut and Limit Shape implementation can be found in https://github.com/ruqihuang/SGP2020_ConsistentZoomOut.

### 4.7 Pearson's r correlation test

To assess the similarity between our Procrustes aligned coordinate-based shape variables and (i) Area-based LSSDs and (ii) Conformal LSSDs we obtained euclidean distance matrices of each representation and conducted Pearson's r correlation tests in R.

### 4.8 *A priori* biological group classification tasks

After obtaining the shape differences through Consistent ZoomOut and Limit Shape, we perform the following classification/clustering tasks to evaluate whether our proposed method is able to extract morphological features that capture and characterize the shape differences present in a collection of hominoid cuboids. In particular, we fit models with six different types of input data. The first two representations of hominoid cuboid shape are based on procrustes aligned landmark configurations of 21 type I-III and 130 sliding semilandmarks, respectively. The third and fourth representations are auto3DGM procrustes pseudolandmarks quantified with -256, and -512 points respectively. The fifth and sixth representations are conformal and area-based LSSDs from our morphVQ analysis. We then calculate the principal components given the six different inputs and choose the number of PCs to cover 95% of the total variance. The PCs are the final inputs for the Genus classification tasks. Specifically, we consider the following four tasks, three of which are supervised classifications and one is based on unsupervised clustering. Namely, we choose Logistic regression, Linear Discriminant Analysis, Naive Bayes as our supervised classifiers. We use K-means to be our unsupervised clustering algorithm. Comparing the performance of PCs with six different inputs on these four algorithms provides insight into which types of the extracted feature are better in identifying morphological differences. Here we provide a brief summary of the four classification (supervised/unsupervised) algorithms:

**4.8.1 Logistic regression.** Logistic regression is a generalized form of linear regression, which regresses the log odds of the probability of choosing one class over the other class. To characterize such correlations, a linear regression model is simply transformed by a sigmoid function. The transformed function models $p(y|x)$, where $x$ refers to the input features (in our case PCs) and $y$ is the class label. Refer to equation for details. Where $a^T$ and $b$ refer to the parameters ("slope" and bias) in the logistic regression.

$$\log\frac{p(y=1|x)}{p(y=0|x)} = a^T x + b \tag{4}$$

**4.8.2 Linear Discriminant Analysis (LDA).** LDA is a linear classifier aimed to find a projection direction such that the input features, once projected with respect to the given direction, have the maximum variance between classes and minimum variance within each of the classes.

**4.8.3 Naive Bayes.** This is one of the simplest classifiers under the assumption that each of the feature dimensions is independent. Taking advantage of Bayes' Rule: $p(y|x) \sim p(x|y)p(y)$,

we further decompose this equation into the multiplication of the independent likelihoods with the probability for the class $y$ : $p(x|y) = \prod_{i=1}^{n}(x_i|y)$, where n denotes the dimension of the inputs. By applying the "maximum a posteriori (MAP)" classification rule, we are able to find the classification given the input features by: $y = \arg\max_k p(y = k)\prod_{i=1}^{n}(x_1|y = k)$, where k denotes the class index.

**4.8.4 K-means.** Different from previous approaches, K-means is an unsupervised approach that separates input data into different clusters based on the distances between each point in the input data and the means of each of the clusters. Here $K$ is a pre-specified hyperparameter that controls the number of clusters in the algorithm. Once it is confirmed (in our case it is the number of species), the algorithm will iteratively update the classifications of points in the input data and the cluster means to minimize the following objective: $\arg\min_S \sum_{i=1}^{k}\sum_{x\in S_i}\| x - \mu_i \|^2$, where $S = S_1, S_2, \ldots, S_k$ and each $S_i$ ($i = 1, 2, \ldots, k$) denotes the set of inputs to class $i$. $i$ refers to the mean of the inputs in $S_i$. And $k$ is a predefined hyperparameter for the number of classes.

To properly evaluate the performance of the above tasks, we apply 11-fold cross validation on the input shape differences. All results are summarized in Table 1 and in S4 Fig with standard deviations indicated according to the cross validation results.

## 4.9 Validation via landmark position estimation accuracy

To validate our approach, we assess the accuracy with which expert-placed ground-truth landmark positions can be determined from auto3DGM and morphVQ representations. We compare discrepancies in the euclidean distances between ground-truth landmark positions and those estimated automatically by both morphVQ and auto3DGM; smaller euclidean distances between true landmarks and estimated landmarks indicate increased accuracy. We placed five landmarks on each mesh in a sample of 69 aligned cuboids. These five landmarks, shown in Fig 10, are the most prominent points on the cuboid identified in [36]. These 69 meshes are common to the 512-pseudolandmark auto3DGM analysis and to our own morphVQ analysis. For each of the two analyses, we then obtain euclidean distance estimates in an iterative fashion for comparison.

For each of the 69 meshes in the auto3DGM analysis, we choose the pseudolandmarks closest in proximity to the five ground-truth landmarks placed on that surface. We then obtain the pseudolandmark positions that correspond to them on all other meshes and measure the euclidean distances between those corresponding pseudolandmarks and the respective ground-truth points. This leaves us with multiple sets of euclidean distances measures (one set for each mesh) that are then retained for comparison.

MorphVQ is designed to estimate functional correspondences. However, landmark positions can be obtained from the resulting P2P maps. For each of the 69 meshes in our morphVQ analysis, we obtain the vertex positions closest in proximity to the five ground-truth landmarks of each surface. Since P2P correspondence between each surface and the other 68 is known, we query the mappings from each surface to all others to obtain estimated landmark positions. We then measure the euclidean distance between those points and the ground-truth points of each mesh. Again, this leaves us with multiple sets of distance measures.

With these sets of distance measurements from MorphVQ and Auto3DGM, we calculate their mean and standard deviations. These two factors are then used to evidence the estimation accuracy of landmark positions between the two approaches.

## Supporting information

**S1 Fig. Harmonic surface network feature extractor used for descriptor learning.** Adapted from Wiersma et. al., 2020. On the left is the U-ResNet structure with three pooling levels, on the right, is a detailed description of the ResNet block.
(TIF)

**S2 Fig. Spectral descriptors learned by our model lead to accurate functional maps.** The first five dimensions of the spectral descriptor learned with our HSN feature extractor. Corresponding/homologous regions on Hylobates source (above) and Pan target (below) shapes are similar in color. These learned descriptors yield high-quality FM correspondences.
(TIF)

**S3 Fig. Correspondence quality is improved by learning descriptors compared to direct optimization with orientation operator.** Point-to-point maps between source *Pan* shape and *Hylobates* target shape (left to right) obtained via direct optimization using WKS (above) and our learned spectral descriptor (below) [25]. Source and target shapes are remeshed to 12,000 vertices in both experiments. WKS were computed using 200 Laplace-Beltrami eigenfunctions.
(TIF)

**S4 Fig. LSSDs lead to similar genus classification accuracies with logistic regression.** Confusion matrices of multinomial logistic regression prediction for each representation of hominoid cuboid shape. In **(A)** and **(B)** use principal components of Procrustes-aligned semilandmark patches and pseudolandmarks generated by auto3DGM as independent variables, respectively. Independent variables in **(C)** and **(D)** are principal components of conformal and area-based LSSDs, respectively. PCs account for 95% shape variance for each representation, and stratified K-Fold cross-validation provides accuracies with standard deviations in parentheses.
(TIF)

## Acknowledgments

We would like to thank the collections managers at the American Museum of Natural History, National Museum of Natural History, Field Museum, and Harvard Museum of Comparative Zoology for access to mammalian osteological collections. We thank Cris Hughes and Lyle Konigsberg for their academic advice and support. We would also like to thank Camille Goudeseune and Travis Ross of the Beckman Institute for their advice and guidance.

## Author Contributions

**Conceptualization:** Oshane O. Thomas.

**Data curation:** Oshane O. Thomas.

**Formal analysis:** Oshane O. Thomas, William E. H. Harcourt-Smith.

**Investigation:** Oshane O. Thomas, Hongyu Shen.

**Methodology:** Oshane O. Thomas, Ryan L. Raaum, Mark Hasegawa-Johnson.

**Project administration:** Oshane O. Thomas.

**Resources:** Oshane O. Thomas, Hongyu Shen.

**Software:** Oshane O. Thomas, Hongyu Shen.

**Supervision:** Ryan L. Raaum, William E. H. Harcourt-Smith, John D. Polk, Mark Hasegawa-Johnson.

**Visualization:** Oshane O. Thomas.

**Writing – original draft:** Oshane O. Thomas.

**Writing – review & editing:** Hongyu Shen, Ryan L. Raaum, William E. H. Harcourt-Smith, John D. Polk, Mark Hasegawa-Johnson.

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
