## [Decision Letter · Decision Letter 0]

10 Jul 2021

Dear Mr. Thomas,

Thank you very much for submitting your manuscript "Automated morphological phenotyping using learned shape descriptors and functional maps: A novel approach to geometric morphometrics" for consideration at PLOS Computational Biology.

As with all papers reviewed by the journal, your manuscript was reviewed by members of the editorial board and by several independent reviewers. In light of the reviews (below this email), we would like to invite the resubmission of a significantly-revised version that takes into account the reviewers' comments.

We cannot make any decision about publication until we have seen the revised manuscript and your response to the reviewers' comments. Your revised manuscript is also likely to be sent to reviewers for further evaluation.

Sincerely,

Min Xu, Phd

Guest Editor

PLOS Computational Biology

Jian Ma

Deputy Editor

PLOS Computational Biology

Reviewer's Responses to Questions

**Comments to the Authors:**

Reviewer #1: This is an ambitious paper that describes a new shape analysis approach that tries to establish functional correspondences between whole triangular meshes (as opposed to landmark configurations, as traditionally done in morphometrics) using descriptor learning. While I find the rationale behind the approach sound, I cannot say that I found this approach successful. My comments to the authors will concentrate on three main issues: generalizability, accuracy, and robustness.

First, starting with the simplest issue: generalizability. The approach in question claims to be a general approach for shape analysis that is aimed at replacing traditional morphometric tools. Yet, the entire manuscript is based on a single dataset, composed of a single morphological structure (102 hominoid cuboids). The structure in question, the cuboid, is also topologically very simple and asymmetric, which greatly facilitates the estimation of shape correspondences. Overall, I find this lacking. At the very least, an effort needs to be made to demonstrate that the approach is indeed general. There is a wealth of triangular meshes freely available online through databases such as Morphosource, Facebase, etc. Skulls, teeth, long bones, etc, represent just a few of the multiple skeletal structures that are abundant in such databases.

Secondly, I think it is worth talking about accuracy. The manuscript makes two attempts to measure accuracy in the predictions. On one side, it looks at species classification accuracy, which I find misleading and indirect (as will be explored in my next point). More importantly, the manuscript also compares manual-automated representations. In my view, this second measurement of accuracy is much more direct and important given the goals of this manuscript. When considering this second measurement of accuracy, I find the approach quite error-prone. As reported on the main text (Page 6), representations derived from the approach presented in this paper are moderately-to-poorly correlated with manual representations (r =0.621). I find this quite low and, in my view, that would be enough to render the approach unusable for practical purposes. While the manuscript attempts to benchmark this in terms of Auto3DGM (claiming it is the only other such approach in the literature), this is factually incorrect, as a large amount of automated approaches became available in recent years (some of which have readily available implementations). To cite a few:

[1] Devine, J., Aponte, J.D., Katz, D.C., Liu, W., Vercio, L.D.L., Forkert, N.D., Marcucio, R., Percival, C.J. and Hallgrímsson, B., 2020. A registration and Deep Learning approach to automated landmark detection for geometric morphometrics. Evolutionary Biology, 47(3), pp.246-259.

[2]Toussaint, N., Redhead, Y., Vidal-García, M., Lo Vercio, L., Liu, W., Fisher, E.M., Hallgrímsson, B., Tybulewicz, V.L., Schnabel, J.A. and Green, J.B., 2021. A landmark-free morphometrics pipeline for high-resolution phenotyping: application to a mouse model of Down syndrome. Development, 148(18), p.dev188631.

[3] Maga, A.M., Tustison, N.J. and Avants, B.B., 2017. A population level atlas of Mus musculus craniofacial skeleton and automated image‐based shape analysis. Journal of anatomy, 231(3), pp.433-443.

[4]Porto, A., Rolfe, S.M. and Maga, A.M., 2020. ALPACA: a fast and accurate approach for automated landmarking of three-dimensional biological structures. bioRxiv.

[5]Frank, Lawrence R., Timothy B. Rowe, Doug M. Boyer, Lawrence M. Witmer, and Vitaly L. Galinsky. "Unveiling the third dimension in morphometry with automated quantitative volumetric computations." (2021).

Finally, my third and last point is related to robustness. In my view, not enough was done to guarantee the robustness of the approach. In particular, the main task being explored in the manuscript is the task of species classification. The manuscript attempts to argue that this approach obtains high classification accuracy (higher than Auto3DGM). I find this misleading, as it is obvious from all the plots (and confusion matrices) that the classification task at hand was extremely “easy”. By “easy”, I mean that species differences are clearly pronounced and, consequently, all methods present very high accuracy levels (0.90 and above). In my view, this classification test tells us little about the robustness of the method.

Altogether, the aims here are laudable, but a number of important issues can be found with the approach. Until a robust test of the method is done, it will be difficult to get behind it.

Reviewer #2: The review has been uploaded as an attached file.

Reviewer #3: Please see the attached document.

Reviewer #4: This paper introduces a new pipeline for morphological analysis for 3D shapes in a fully automatic way using the functional map framework. Rather than relying on manually specified landmarks, the authors propose to use a fully automatic correspondence estimation pipeline that was developed in the geometry processing/vision communities and demonstrate its use in geometric morphometrics. The main novelty of this paper, therefore, is to demonstrate the applicability of this automatic pipeline for both phenotyping and for highlighting areas of variability. The main application is for analyzing hominoid cuboids, and using both the automatically detected correspondences and the LSSDs (which capture shape variability in a collection of shapes linked by functional maps) for fully automatic morphological phenotyping.

I am not an expert in geometric morphometrics (my expertise is mainly in 3D shape analysis), but I thought both the motivation and the technical execution of the paper were convincing. The authors have done their due diligence in investigating and adapting the latest relevant developments in the functional map literature to their own problem setting and have demonstrated a compelling evaluation compared to baseline approaches, both manual and automatic. I particularly appreciated the use of unsupervised learning for computing features by adapting the SURFMNet architecture to the given shape collection, while using the HSN feature extractor. Furthermore, the authors used the very recent consistent zoomout and Latent Space Shape Differences (LSSDs) to encode shape variability. These are non-trivial concepts that are all nicely brought together in this work.

I have a few relatively minor comments and suggestions:

1. The authors mention that their shapes should be approximately rigidly aligned ("Our algorithm performs best when polygon models are approximately rigidly aligned..."). However, later on the authors also mention that they adopt the HSN feature extractor because it is capable of producing rotationally-invariant features. "This is usually the case for real-world pairs of bone meshes which may be poorly aligned, if at all." These two statements seem contradictory -- if the shapes need to be pre-aligned anyway, what is the advantage of using HSN? Furthermore, in some functional map learning frameworks such as [60], data augmentation is used to achieve rotation invariance. I think it's important to comment on exactly how much rotation invariance is needed and where it plays most role.

2. The functional map pipeline has recently been used for morphometric analysis in "Geometric analysis of shape variability of lower jaws of prehistoric humans" by Ren et al. That work should be cited and discussed at least in some detail, as it seems to tackle similar goals as presented in this submission. Furthermore, even the code of that work is publicly available at https://github.com/llorz/PROJ2020_jaws Therefore, it would be reasonable to expect a comparison to that work as another baseline. It would be interesting and useful to know which sets of approaches are most accurate and robust.

3. Some citations seem to be not properly formatted. For example, [62] is missing the venue. The same is true for many other references, such as [58], [21] and others. Furthermore, some titles are wrong such as "A Functional View of Geometry Processing" should both include a venue and should not have an article at the beginning. A careful proof-reading of the bibliography and fixing all the references would be useful.

4. The authors rely on the Latent Space Shape Differences, which seem to arise from a paper that is only on arxiv. Instead another paper "Limit shapes–a tool for understanding shape differences and variability in 3d model collections" published by the same authors has appeared in the Computer Graphics Forum. In that paper the authors define Characteristic Shape Differences, which correspond to the same concept. If there is any distinction between these concepts, it would be useful to state it explicitly. If not, I would suggest the authors to use the terminology from the more recent, published work.

Despite these minor concerns, I think this paper makes a valuable contribution, both by adapting the functional map framework to a novel domain and by introducing a fully automatic morphological phenotyping technique. I therefore believe that this paper is likely to inspire future work.

Finally, I have not checked but it would be very useful if the authors released their data and code in order to facilitate further research in this area.

**Have the authors made all data and (if applicable) computational code underlying the findings in their manuscript fully available?**

Reviewer #1: **No: **The data availability statement indicates the data will be deposited in the future.

Reviewer #2: **No: **The authors did not provide the mesh models, landmark coordinates, and FMs used to obtain the results.

Reviewer #3: Yes

Reviewer #4: **No: **I have not checked -- but I did not receive any information about whether the data and/or code were submitted, and if so, how one can access them.

PLOS authors have the option to publish the peer review history of their article (what does this mean?). If published, this will include your full peer review and any attached files.

Reviewer #1: No

Reviewer #2: No

Reviewer #3: No

Reviewer #4: No
---

## [Decision Letter · Decision Letter 1]

25 Apr 2022

Dear Mr. Thomas,

Thank you very much for submitting your revised manuscript "Automated morphological phenotyping using learned shape descriptors and functional maps: A novel approach to geometric morphometrics" for consideration at PLOS Computational Biology.

As with all papers reviewed by the journal, your revised manuscript was reviewed by members of the editorial board and by the independent reviewers. In light of the reviews (below this email), we would like to invite the resubmission of a revised version that takes into account the reviewers' comments.

We cannot make any decision about publication until we have seen the further revised manuscript and your response to the reviewers' comments. Your further revised manuscript is also likely to be sent to reviewers for additional evaluation.

[1] A letter containing a detailed list of your responses to the comments to your latest revision, and a description of the changes you have made in the manuscript. Please note while forming your response, if your article is accepted, you may have the opportunity to make the peer review history publicly available. The record will include editor decision letters (with reviews) and your responses to reviewer comments. If eligible, we will contact you to opt in or out.

Sincerely,

Min Xu, Phd

Guest Editor

PLOS Computational Biology

Jian Ma

Deputy Editor

PLOS Computational Biology

Reviewer's Responses to Questions

**Comments to the Authors:**

Reviewer #1: The authors have made a commendable effort in replying to prior criticism. In general, I do find the approach ingenious and its comparison to Auto3DGM appropriate. I think, however, that some of my prior concerns have not been truly addressed.

In particular, I still think this manuscript relies on topologically simple skeletal structures, in which homology is hard to define. Another way of putting this would be: how will this method fare with a structure such as the mammalian skull, where sutures delineate sharp homology boundaries? My intuition is that the current approach would not reflect any notion of biological homology, despite reflecting general geometric similarity. That is the reason why I emphasized landmark locations in the previous iteration. I do not think landmarks themselves are a better approach for describing shape variation. I simply think that the manuscript needs to clearly evaluate the extent to which these geometric correspondences reflect any notion of biological homology. I still cannot answer that question, given the current version of the manuscript.

Having said that, I do understand that I am mostly alone in my concerns, with other reviewer's being mostly positive about the manuscript. I sincerely hope, however, that the authors will take these concerns seriously.

Reviewer #2: I went over the revised version of the manuscript and the authors rebuttable to my previous comments. Indeed, it appears that all of my most major concerns have been adequately addressed, while a sufficient explanation was provided for those which were not. Therefore, I recommend the paper to be published in POLS computational biology.

Reviewer #3: My review is uploaded as an attachment.

Reviewer #4: This is a revision of the paper that I reviewed previously. The authors have incorporated the majority of all of the required changes and have substantially revised the manuscript. The cover letter and summary of changes were particularly useful. The main changes with respect to the previous version include demonstrating the performance of the proposed approach on two other datasets analyzing shape variation in mouse humeri and in primate medial cuneiforms. This substantially improves the scope and impact of the proposed method and highlights its general power.

Overall, I find this paper of significant value as it aims to address a fundamental problem in Geometric Morphometrics, reducing the need for manual intervention and helping to avoid annotation bias. As such, I think it can open the door to future works and constitutes an important contribution to the field.

When preparing the final version, I only have two suggestions for the authors:

1. I would like to request a more detailed discussion about the importance of rotation invariance in the analysis process. Specifically, I did not quite understand the response provided by the authors in the cover letter about this question. "Rotation-invariance" is a technical term that means that the feature extractor should produce the same exact features regardless of the orientation in space. It seems that the authors advocate the use HSN because it is *robust* with respect to small orientation changes. The authors mention "However, this HSN structure is still able to have some level of rotation invariance (e.g. +- 5 degrees in our experiments)." This to me is not quite clear, since a rotationally-invariant feature extractor should produce exactly the same features regardless of orientation. The sentence by the authors, rather, suggests "robustness" with respect to small orientation changes, instead of true rotational "invariance." I think this should be clarified.

2. Since the authors emphasize the efficiency of their approach, it would be worthwhile to add timings for different stages of their approach. Perhaps I missed it, but I could not a detailed timing report. In the cover letter the authors mention "An Auto3DGM analysis of the same shape collection with 512 pseudolandmarks took 26 days to complete, our pipeline took 2 days. " This definitely has to be reported in the manuscript and also I would suggest to report the processing cost of different stages in the algorithm to see what parts take the most amount of time. This could help to appreciate the efficacy and bottlenecks of the proposed approach.

Apart from these minor comments, I think the manuscript is in great shape and I would be happy to recommend its acceptance.

**Have the authors made all data and (if applicable) computational code underlying the findings in their manuscript fully available?**

Reviewer #1: **No: **Have not found the dryad repository in question

Reviewer #2: Yes

Reviewer #3: Yes

Reviewer #4: **No: **I did not check in detail, but I did not see a complete data+code release

PLOS authors have the option to publish the peer review history of their article (what does this mean?). If published, this will include your full peer review and any attached files.

Reviewer #1: No

Reviewer #2: No

Reviewer #3: No

Reviewer #4: No
---

## [Decision Letter · Decision Letter 2]

24 Aug 2022

Dear Mr. Thomas,

Thank you very much for submitting your manuscript "Automated morphological phenotyping using learned shape descriptors and functional maps: A novel approach to geometric morphometrics" for consideration at PLOS Computational Biology. As with all papers reviewed by the journal, your manuscript was reviewed by members of the editorial board and by several independent reviewers. The reviewers appreciated the attention to an important topic.

Based on the reviews, we are likely to accept this manuscript for publication, providing that you modify the manuscript according to the review recommendations. In particular, we agree with the reviewers that it is imperative that the you make the code and data publicly available..

Sincerely,

Min Xu, Phd

Guest Editor

PLOS Computational Biology

Jian Ma

Section Editor

PLOS Computational Biology

[LINK]

According to reviewer 1, it is encouraged that the authors to make the code and data available to public or at least to the reviewers.

Reviewer's Responses to Questions

**Comments to the Authors:**

Reviewer #1: I am generally satisfied with the revisions. I would like to thank the authors for engaging with my prior comments. However, the code is not available on the github repository (https://github.com/oothomas/morphVQ). Similarly, I could not find the data corresponding to the manuscript on Dryad.

**Have the authors made all data and (if applicable) computational code underlying the findings in their manuscript fully available?**

Reviewer #1: **No: **The code is not available on the github repository (https://github.com/oothomas/morphVQ). Similarly, I could not find the data corresponding to the manuscript on Dryad.

PLOS authors have the option to publish the peer review history of their article (what does this mean?). If published, this will include your full peer review and any attached files.

Reviewer #1: No

Figure Files:

Data Requirements:

Reproducibility:

References:

---

## [Editor Report · Decision Letter 3]

13 Nov 2022

Dear Mr. Thomas,

We are pleased to inform you that your manuscript 'Automated morphological phenotyping using learned shape descriptors and functional maps: A novel approach to geometric morphometrics' has been provisionally accepted for publication in PLOS Computational Biology.

Best regards,

Min Xu, Phd

Guest Editor

PLOS Computational Biology

Jian Ma

Section Editor

PLOS Computational Biology

---

## [Editor Report · Acceptance letter]

12 Jan 2023

PCOMPBIOL-D-21-00802R3 

Automated morphological phenotyping using learned shape descriptors and functional maps: A novel approach to geometric morphometrics

Dear Dr Thomas,

I am pleased to inform you that your manuscript has been formally accepted for publication in PLOS Computational Biology. Your manuscript is now with our production department and you will be notified of the publication date in due course.

With kind regards,

Anita Estes
